# A Novel 16S rRNA PCR-Restriction Fragment Length Polymorphism Assay to Accurately Distinguish Zoonotic *Capnocytophaga canimorsus* and *C. cynodegmi*

Cheng-Hung Lai,[a,b] Yu-Sin Lin,[a] (iD)Chao-Min Wang,[c] Poa-Chun Chang,[d] (iD)Wei-Yau Shia[a,b]

[a]Department of Veterinary Medicine, College of Veterinary Medicine, National Chung Hsing University, Taichung, Taiwan
[b]Veterinary Medical Teaching Hospital, College of Veterinary Medicine, National Chung Hsing University, Taichung, Taiwan
[c]Department of Veterinary Medicine, National Chiayi University, Chiayi City, Taiwan
[d]Graduate Institute of Microbiology and Public Health, National Chung Hsing University, Taichung, Taiwan

**ABSTRACT** The zoonotic bacteria *Capnocytophaga canimorsus* and *C. cynodegmi*, the predominant *Capnocytophaga* species in the canine oral biota, can cause human local wound infections or lethal sepsis, usually transmitted through dog bites. Molecular surveying of these *Capnocytophaga* species using conventional 16S rRNA-based PCR is not always accurate due to their high genetic homogeneity. In this study, we isolated *Capnocytophaga* spp. from the canine oral cavity and identified them using 16S rRNA and phylogenetic analysis. A novel 16S rRNA PCR-restriction fragment length polymorphism (RFLP) method was designed based on our isolates and validated using published *C. canimorsus* and *C. cynodegmi* 16S rRNA sequences. The results showed that 51% of dogs carried *Capnocytophaga* spp. Among these, *C. cynodegmi* (47/98, 48%) was the predominant isolated species along with one strain of *C. canimorsus* (1/98, 1%). Alignment analysis of 16S rRNA sequences revealed specific site nucleotide diversity in 23% (11/47) of the *C. cynodegmi* isolates, which were misidentified as *C. canimorsus* using previously reported species-specific PCR. Four RFLP types could be classified from all the isolated *Capnocytophaga* strains. The proposed method demonstrates superior resolution in distinguishing *C. cynodegmi* (with site-specific polymorphism) from *C. canimorsus* and especially in distinguishing *C. canimorsus* from other *Capnocytophaga* species. After *in silico* validation, this method was revealed to have an overall detection accuracy of 84%; notably, accuracy reached 100% in *C. canimorsus* strains isolated from human patients. Overall, the proposed method is a useful molecular tool for the epidemiological study of *Capnocytophaga* in small animals and for the rapid diagnosis of human *C. canimorsus* infections.

**IMPORTANCE** With the increased number of small animal breeding populations, zoonotic infections associated with small animals need to be taken more seriously. *Capnocytophaga canimorsus* and *C. cynodegmi* are part of common biota in the mouths of small animals and can cause human infections through bites or scratches. In this study, *C. cynodegmi* with site-specific 16S rRNA sequence polymorphisms was erroneously identified as *C. canimorsus* during the investigation of canine *Capnocytophaga* by conventional PCR. Consequently, the prevalence of *C. canimorsus* is incorrectly overestimated in epidemiological studies in small animals. We designed a new 16S rRNA PCR-RFLP method to accurately distinguish zoonotic *C. canimorsus* from *C. cynodegmi*. After validation against published *Capnocytophaga* strains, this novel molecular method had high accuracy and could detect 100% of *C. canimorsus*-strain infections in humans. This novel method can be used for epidemiological studies and the diagnosis of human *Capnocytophaga* infection following exposure to small animals.

**KEYWORDS** *Capnocytophaga*, phylogenetic analysis, dogs, PCR-restriction fragment length polymorphism, zoonosis

**Ad Hoc Peer Reviewers** (iD)Tintu Abraham, Jawaharlal Institute of Post Graduate Medical Education and Research; (iD)Özlem Şahan Yapıcıer, Mehmet Akif Ersoy University
Address correspondence to Wei-Yau Shia, vmwyshia@nchu.edu.tw.
The authors declare no conflict of interest.

The genus *Capnocytophaga* comprises fastidious, Gram-negative, thin or filamentous rods with tapered or spindle-shaped ends, and are facultative anaerobic and capnophilic bacteria (1, 2). Four *Capnocytophaga* species have been isolated from the canine oral cavity; among these, *C. canimorsus* and *C. cynodegmi* are more predominant than *C. canis* and *C. stomatis* (3–6). They are most commonly transmitted to humans via dog bites, followed by scratches or close contact (7). Although infected patients exhibit symptoms ranging from mild to fulminant, immunocompromised patients tend to show more severe clinical signs (7, 8). Compared to other species, *C. canimorsus* usually causes systemic infections, including cellulitis, meningitis, and sepsis (9). *C. cynodegmi* is considered less pathogenic than *C. canimorsus*, as most *C. cynodegmi*-infected cases develop local wound infection (cellulitis) which rarely advances to systemic infection (10, 11).

In some epidemiological studies, culture- and/or direct PCR-based techniques are often used to evaluate the potential hazards of zoonotic *Capnocytophaga* spp. transmission from the canine oral cavity to humans. Among the species investigated, *C. canimorsus* has been the most extensively surveyed (3, 5, 12–15). PCR-based methods usually reveal a higher prevalence of *Capnocytophaga* than culture-based methods because of the fastidious property of this genus and the high sensitivity of PCR. However, without the culture process, the phenotypic and other important genotypic characteristics of *Capnocytophaga* cannot be determined.

Genetic identification of *Capnocytophaga* spp. is commonly performed by 16S rRNA sequencing (3, 15). However, considering the high homogeneity of the 16S rRNA gene between *C. canimorsus* and *C. cynodegmi*, other techniques, including pulsed-field gel electrophoresis (PFGE), multilocus sequence typing (MLST), and PCR-restriction fragment length polymorphism (RFLP), have also been applied to discriminate between canine *Capnocytophaga* spp. (14–17). Suzuki et al. developed genus- and species-specific PCR targeting the 16S rRNA gene to allow rapid identification and distinction between *C. canimorsus* and *C. cynodegmi* strains isolated from dogs and cats (5). These primer sets have also been widely adopted in other studies to investigate the prevalence of *Capnocytophaga* spp. in small animals (4, 14, 18).

In this study, we discovered that the 16S rRNA sequence polymorphism at the specific location in *C. cynodegmi* strains showed ambiguous results when using the previously established species-specific PCR for culture-based epidemiological investigation of *Capnocytophaga* in dogs. Therefore, we utilized the isolated strains to develop a novel 16S rRNA PCR-RFLP method which accurately distinguished between *C. canimorsus* and *C. cynodegmi*, and demonstrated its potential application in future epidemiological studies and disease diagnosis.

## RESULTS

**Prevalence investigated using conventional methods: species-specific PCR and 16S rRNA sequences.** Oral swabs were collected from 98 dogs (50 males and 48 females), including 82 owned and 16 sheltered dogs. The median age of the dogs was 8 years (range: 0.25 to 17 years). Relevant individual parameters were summarized; none of the basic individual parameters was significantly correlated with the presence of *Capnocytophaga* (Table S1). The prevalence of *Capnocytophaga* sp. in the canine oral cavity was 51% (49/98). Fifty *Capnocytophaga* strains were isolated from 49 dogs. After sequencing the amplified 16S rRNA genes of these 50 strains and trimming the poor signal sequences from both ends, sequences of approximately 1,331 bp were used for the BLAST and subsequent phylogenetic analysis. *C. cynodegmi* showed the same high prevalence (47.96%, 47/98) among all the sampled dogs in both the species-specific PCR and 16S rRNA sequence BLAST results. The differences in the samples surveyed by species-specific PCR and 16S rRNA sequencing in *C. canimorsus* were 12.24% (12/98) and 1.02% (1/98), respectively. Eleven strains identified as *C. cynodegmi* based on the 16S rRNA gene, with identities ranging from 98.5% to 99.7%, were positive for both *C. canimorsus* and *C. cynodegmi* by species-specific PCR. The other two strains (62067N and 91462) were identified as *C. cynodegmi* based on the 16S rRNA gene, with 97.4% and 95.7% identity, respectively. These are likely to be new species and remained

A

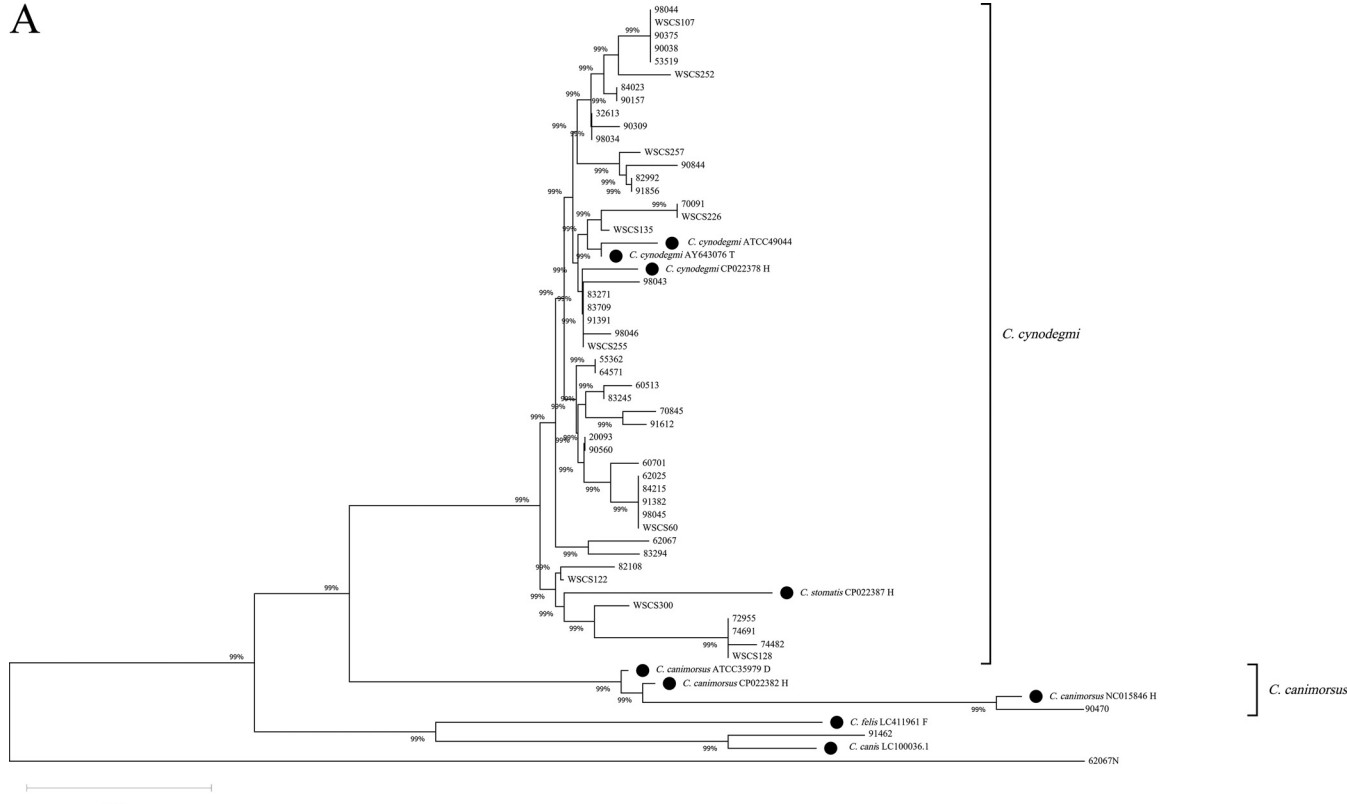

**FIG 1** Phylogenetic tree and molecular typing results according to the 16S rRNA gene sequences of *Capnocytophaga* isolates and reference strains. (A) Phylogenetic tree of all *Capnocytophaga* isolates and reference strains. Reference strains are indicated by filled circles. Published sequences are presented as strain no., GenBank accession no., and source. (B) Molecular typing results of each isolate are shown in a 16S rRNA phylogenetic tree. Black squares and white squares correspond to strains that were positive and negative, respectively, in the catalase test, oxidase test, or *C. canimorsus*-specific (CaR) PCR test. Molecular typing according to the nucleotide polymorphism of species-specific PCR reverse primer binding sites is presented using a color code, with the key shown in the left upper corner of the diagram. T, type strain; H, human isolate; D, canine isolate; F, feline isolate.

unidentified in the genus- and species-specific PCR. Two different *Capnocytophaga* species were present in a single specimen: *C. cynodegmi* and a suspected new species (strain 62067N).

**Effect of site-specific sequence polymorphism on genus- and species-specific PCR and the phylogenetic tree.** We constructed a phylogenetic tree based on the 16S rRNA sequences of the 50 isolates with characteristic annotation and 9 reference strains obtained from GenBank (Fig. 1). The only *C. canimorsus* isolate was well separated from the *C. cynodegmi* clade, with 99% bootstrap support. In contrast, the 47 *C. cynodegmi* isolates showed high similarity. Strain 91462 belonged to the same clade as *C. canis*, with 99% bootstrap support; however, the 16S rRNA BLAST results showed 97.4% identity with *C. cynodegmi*, suggesting the need for further species identification. Another unidentified *Capnocytophaga* strain (62067N) was shown to be entirely independent of other canine-related *Capnocytophaga* strains (Fig. 1A). The reference strain *C. stomatis* H2177 (GenBank accession no. NZ_CP022387.1) was not well distinguished from *C. cynodegmi* according to the 16S rRNA-based phylogenetic tree.

The aforementioned 13 *Capnocytophaga* isolates showed inconsistent results between the 16S rRNA gene BLAST and genus- and species-specific PCR. The negative PCR results (strains 91462 and 62067N) are likely due to marked differences in the recognition sequences of the forward primer CaL2 shared by genus- and species-specific PCR (Fig. 2A and B). Upon further analysis of *C. cynodegmi*-specific PCR reverse primer recognition sequences in our isolates, the 50 *Capnocytophaga* strains were classified into five different types, A to E (Fig. 1B and 2C). Strains belonging to types A, C, D, and E were consistent with the species-specific PCR results and 16S rRNA gene sequence BLAST analysis. The type B strains, mostly aggregated in two clades, were all identified as *C. cynodegmi*—the 11 strains described

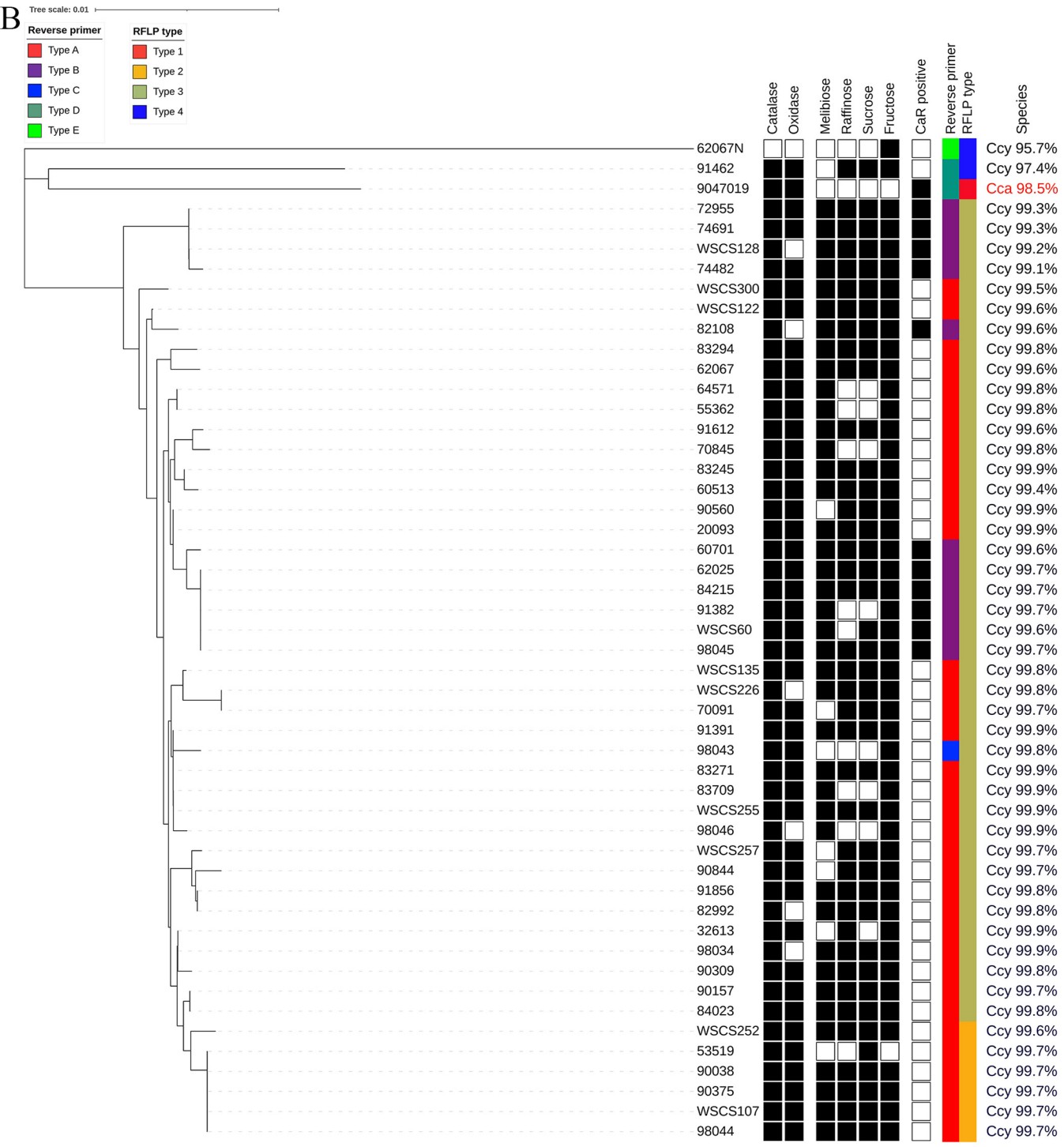

**FIG 1** (Continued)

above which exhibited both *C. canimorsus*- and *C. cynodegmi*-specific PCR positivity (Fig. 1B). These strains showed a substitution of nucleotide G for A at 16S rRNA gene position 482, which resulted in only two nucleotides (positions 479 and 483) differing from the *C. canimorsus*-specific reverse primer (Fig. 2C). This explains the ambiguous species-specific PCR results of the type B strains.

**Biochemical identification.** From the biochemical profiling, only three strains (3/47) had species IDs matching their 16S rRNA results. No ID was shown for the other 44 isolates, which showed low similarity indices. Among these isolates, the results for 82% (36/44) of the strains were consistent with the 16S rRNA sequencing results. In contrast,

## A

### Position number according to L14637

| Strain | 175 | 176 | 177 | 178 | 179 | 180 | 181 | 182 | 183 | 184 | 185 | 186 | 187 | 188 | 189 | 190 | 191 | 192 | 193 | 194 | 195 | 196 |
|---|---|---|---|---|---|---|---|---|---|---|---|---|---|---|---|---|---|---|---|---|---|---|
| Primer AS1 | T | A | G | T | A | T | T | G | T | T | T | G | G | T | G | G | C | A | T | C | A | C |
| Cca ATCC 35979 | . | . | . | . | . | . | . | . | . | . | . | . | . | . | . | . | . | . | . | . | . | . |
| Ccy ATCC 49044 | . | . | . | . | . | . | . | . | . | . | . | . | . | . | . | . | . | . | . | . | . | . |
| 90470 | . | . | . | . | . | . | . | . | . | . | . | . | . | . | . | . | . | . | . | . | . | . |
| 70091 | . | . | . | . | . | . | . | A | . | . | . | A | . | . | . | . | . | . | . | . | . | T |
| 20093 | . | . | . | . | . | . | . | A | . | . | . | . | . | . | . | . | . | . | . | . | . | T |
| 91462 | . | . | . | . | . | . | . | A | . | . | T | A | . | . | . | . | . | . | . | . | . | T |
| 62067N | . | . | . | . | . | . | . | A | . | . | T | . | . | . | . | . | . | . | . | . | . | T |

## B

### Position number according to L14637

| Strain | 71 | 72 | 73 | | | 74 | 75 | 76 | 77 | 78 | 79 | 80 | 81 | 82 | 83 | 84 | 85 | 86 | 87 | 88 | 89 | 90 |
|---|---|---|---|---|---|---|---|---|---|---|---|---|---|---|---|---|---|---|---|---|---|---|
| Primer CaL2 | G | T | A | - | - | G | A | G | T | G | C | T | T | C | G | G | C | A | C | T | T | G |
| Cca ATCC 35979 | . | . | . | . | . | . | . | . | . | . | . | . | . | . | . | . | . | . | . | . | . | . |
| Ccy ATCC 49044 | . | . | . | . | . | . | . | . | . | . | . | . | . | . | . | . | . | . | . | . | . | . |
| 70091 | . | . | . | . | . | . | . | . | . | . | . | . | . | . | A | . | . | . | . | . | . | . |
| 98043 | . | . | . | . | . | . | . | . | . | . | . | . | . | . | . | . | . | G | . | . | . | . |
| WSCS300 | . | . | . | . | . | . | . | . | . | . | T | . | . | . | . | . | . | . | . | . | . | . |
| 60701 | . | . | . | . | . | . | G | . | . | . | . | . | . | . | . | . | . | G | . | . | . | . |
| 20093 | . | . | . | . | . | . | G | . | . | . | . | . | . | . | . | . | . | . | . | . | . | . |
| 91462 | . | . | . | A | G | . | . | . | A | . | . | . | . | G | C | T | . | T | . | C | C | T |
| 62067N | . | . | . | . | . | A | G | C | . | A | . | . | . | . | . | T | . | G | . | C | C | T |

## C

### Position number according to L14637

| Group | 479 | 480 | 481 | 482 | 483 | 484 | 485 | 486 | 487 | 488 | 489 | 490 | 491 | 492 | 493 | 494 | 495 | 496 | 497 |
|---|---|---|---|---|---|---|---|---|---|---|---|---|---|---|---|---|---|---|---|
| Primer CyR | C | A | T | A | C | G | A | A | T | A | A | G | C | A | T | C | G | G | C |
| Primer CaR | T | G | . | . | T | . | . | . | . | . | . | . | . | . | . | . | . | . | . |
| Ccy ATCC 49044 | . | . | . | . | . | . | . | . | . | . | . | . | . | . | . | . | . | . | . |
| Cca ATCC 35979 | T | G | . | . | T | . | . | . | . | . | . | . | . | . | . | . | . | . | . |
| Type | | | | | | | | | | | | | | | | | | | |
| Type A | . | . | . | . | . | . | . | . | . | . | . | . | . | . | . | . | . | . | . |
| Type B | . | G | . | . | . | . | . | . | . | . | . | . | . | . | . | . | . | . | . |
| Type C | . | . | . | G | . | . | . | . | . | . | . | . | . | . | . | . | . | . | . |
| Type D | T | G | . | . | T | . | . | . | . | . | . | . | . | . | . | . | . | . | . |
| Type E | T | C | . | G | . | . | . | . | . | . | . | . | . | . | . | . | . | . | . |

**FIG 2** Nucleotide diversity of genus- or species-specific PCR primer recognition sites among 50 *Capnocytophaga* isolates. Five types were subdivided according to nucleotide polymorphisms from positions 479 to 483. (A) Sequences aligned with the genus-specific PCR forward primer. (B) Sequences aligned with the genus-specific PCR reverse primer. (C) Sequences aligned with the species-specific PCR reverse primer. Gray boxes indicate nucleotides that differ between the two reverse primers of the species-specific PCR.

18% (8/44) of the *C. cynodegmi* strains were misidentified as *C. canimorsus* (5 strains) or other non-*Capnocytophaga* species (3 strains).

Specific carbohydrate utilization (previously used to differentiate *C. canimorsus* from *C. cynodegmi*) was applied to our isolates, and the results are shown in Fig. 1B. All 47 strains of *C. cynodegmi* were catalase positive, and 87.2% (41/47) were oxidase positive (Fig. 1B). The single *C. canimorsus* strain showed positive results for both the catalase and oxidase tests. The unidentified *Capnocytophaga* strain (62067N) was the only strain negative for both the catalase and oxidase tests.

**Differentiation of *C. canimorsus* and C. *cynodegmi* using the novel 16S rRNA PCR-RFLP method.** The amplified 16S rRNA gene products of 50 *Capnocytophaga* isolates were treated with REs, KpnI, and PpuMI and analyzed using gel electrophoresis (Fig. 3). Only the PCR product from *C. canimorsus* was cleaved by PpuMI. In contrast, the PCR products of *C. cynodegmi* and the two unidentified strains (91462 and 62067N) were cleaved only by KpnI at different sites. Using the 16S rRNA gene of the reference strain, four patterns were predicted *in silico* by combining KpnI and PpuMI (Table 1 and Fig. 1B). The type 1 pattern was represented by fragment patterns of 14, 51, 286, and 1,114 bp in the *C. canimorsus* reference strains; our isolates showed consistent results (Fig. 3). However, the small molecular weight fragments (<100 bp) are barely visible because of poor stainability. The *C. cynodegmi* isolates showed two different restriction fragment patterns after treatment with KpnI: the type 2 pattern (478 and 987 bp) was found in 6 isolates, whereas the type 3 pattern (51, 478, and 936 bp) was found in 41 isolates and three reference strains. As predicted *in silico* (Table 1), the reference strain *C. stomatis* showed the same pattern as *C. cynodegmi* (type 3). The last pattern, type 4 (51 and 1,414 bp), was observed in the reference strains of *C. felis* and *C. canis*, and in our unidentified strains (Table 1). In summary, *C. canimorsus* (fragments 286 and 1,114 bp) and *C. cynodegmi* (fragments 478 and 936 bp or 987 bp) could be clearly and easily differentiated by gel electrophoresis of the 16S rRNA PCR amplicons digested by KpnI and PpuMI (Fig. 3).

The RFLP type could be predicted in combination with site-specific nucleotide polymorphisms. Based on the KpnI cleavage site, sequences with a "C" nucleotide at position 479 can be cleaved by KpnI at position 480 (Fig. 2C). Therefore, the reverse primer types A, B, and C correspond to RFLP type 2 or 3, while types D and E correspond to RFLP type 1 or 4 (Fig. 1B).

**Verification of expected RE sites in the published *C. canimorsus* and C. *cynodegmi* strains.** After phylogenetic analysis, 172 *C. canimorsus* and 35 *C. cynodegmi* 16S rRNA gene sequences were collected. The sequence of the expected restriction enzyme (RE) site of each strain is presented in the supplemental material (Fig. S1 and S2). The results showed that 83% (142/172) of *C. canimorsus* and 91% (32/35) of *C. cynodegmi* strains were consistent with the designed 16S rRNA PCR-RFLP results. The overall accuracy for RE detection was 84% (174/207). Importantly, the accuracy for detecting *C. canimorsus* strains isolated from human patients was 100%. All the strains (30 *C. canimorsus* and 3 *C. cynodegmi* strains) that showed inconsistent results (no detection of the predicted restriction sites) were isolated from dogs and cats. Failure to identify the expected RE in *C. canimorsus* strains mainly (28/30, 93.3%) resulted from a C/T substitution at position 289 and a G/T substitution at position 300. The other two *C. canimorsus* strains (accession no. GQ167582.1 and GQ167602.1) did not show the expected RE due to a single deletion at the expected RE position (position 288). A similar finding was obtained in three *C. cynodegmi* strains, where the deletion occurred at position 475 (Fig. S1 and S2).

The phylogenetic tree indicated that 28 of the 30 *C. canimorsus* strains lacking the expected RE site were allocated to a separate clade, with 89% bootstrap support (Fig. 4A). In contrast, the two *C. canimorsus* strains with a single deletion mentioned above (accession no. GQ167582.1 and GQ167602.1) were phylogenetically close to other *C. canimorsus* strains (Fig. 4A). Similarly, three strains of *C. cynodegmi* with a single deletion at the expected RE site were in proximity to the neighboring clades of other *C. cynodegmi* strains (Fig. 4B).

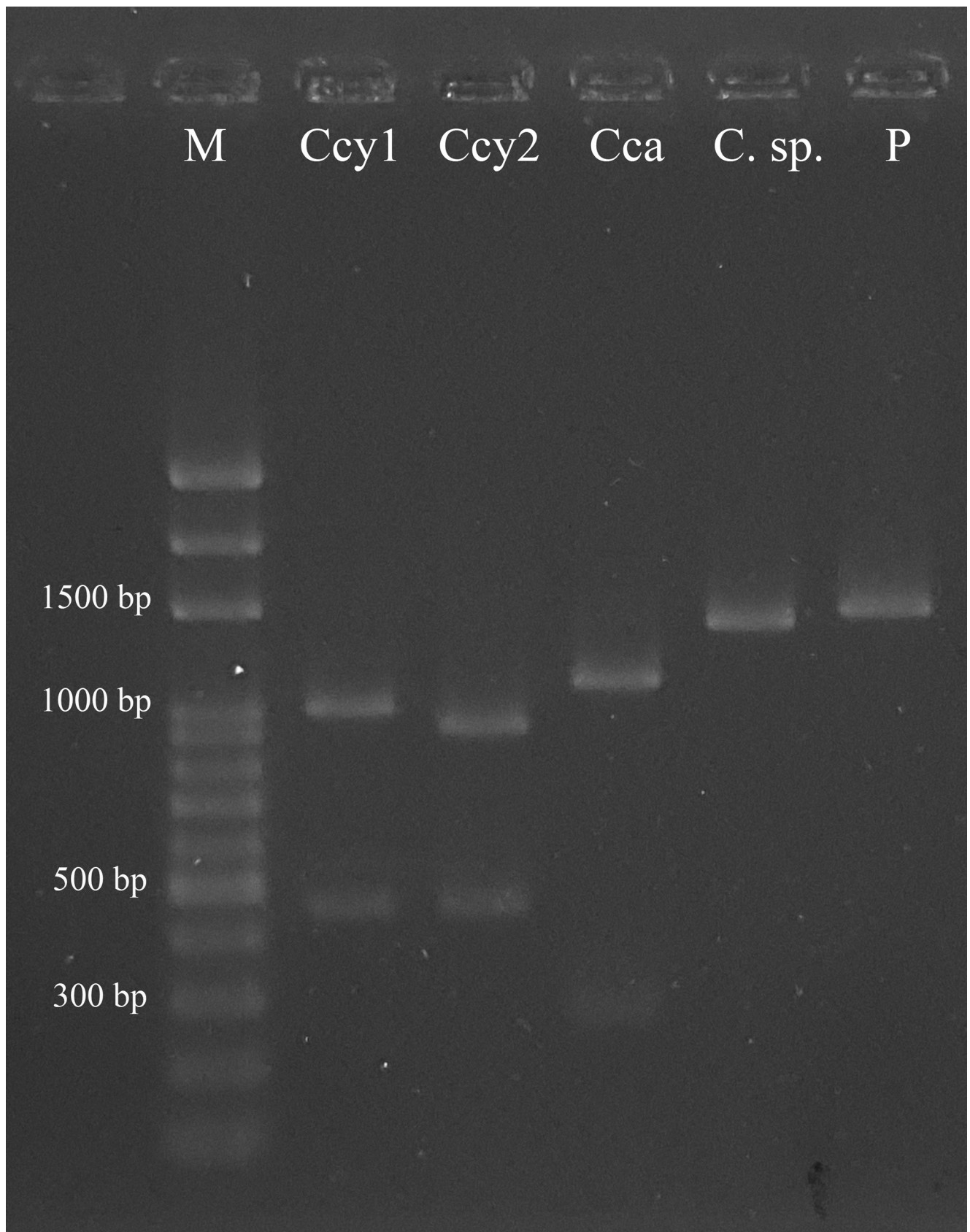

**FIG 3** PpuMI and KpnI restriction patterns for the digestion of amplified 16S rRNA gene products. DNA fragments were analyzed by 1.5% agarose gel electrophoresis. Small molecular weight fragments (<100 bp) are unrecognizable in the image. M, 100-bp marker; Ccy1, *C. cynodegmi* belonging to RFLP

**TABLE 1** Newly designed 16S rRNA PCR-RFLP for identifying the *Capnocytophaga* species and isolates obtained in this study

| | RE cutting site[a] | | | | | | |
|---|---|---|---|---|---|---|---|
| | PpuMI | | KpnI | | | | |
| Pattern type | 286 | 300 | 478 | 1,414 | Fragment sizes (bp) | Represented species | No. of isolates (species) |
| 1 | + | + | − | + | 14, 51, 286, 1,114 | *C. canimorsus* | 1 (*C. canimorsus*) |
| 2 | − | − | + | − | 478, 987 | *C. cynodegmi* | 6 (*C. cynodegmi*) |
| 3 | − | − | + | + | 51, 478, 936 | *C. cynodegmi, C. stomatis* | 41 (*C. cynodegmi*) |
| 4 | − | − | − | + | 51, 1,414 | *C. felis, C. canis* | 2 suspected new species |

[a]Position number (bp) according to L14637. RE, restriction enzyme.

## DISCUSSION

In this study, we investigated the presence of *Capnocytophaga* in the canine oral cavity in Taiwan using a novel combination of culture-based and molecular methods. Molecular screening of *Capnocytophaga* using DNA extracted directly from specimens allows rapid identification in large-scale studies; however, the absence of a culture method reduces the specificity and accuracy of molecular methods. A combination of culture-based and molecular methods facilitates the realistic characterization of a specimen. In this study, we discovered some *C. cynodegmi* isolates with nucleotide polymorphisms that were poorly detected using the existing species-specific PCR methods; however, these strains were correctly identified using our newly developed 16S rRNA PCR-RFLP.

The molecular techniques used to identify and differentiate *C. canimorsus*- and *C. cynodegmi*-related species are summarized in Table 2. Similar PCR-RFLP methods have been reported to identify the presence of *Capnocytophaga* spp. and distinguish between *C. canimorsus* and *C. cynodegmi* (15, 16). Ciantar et al. (16) used CfoI digestion of PCR-amplified targeted 16S rRNA gene fragments to successfully distinguish between human isolate-derived *Capnocytophaga* species. When we applied this method to our isolates, *C. canimorsus* and *C. cynodegmi* only differed by a fragment of 100 bp, which is not easily identified because of its low molecular weight. Another study utilized two REs (StyI and MseI) to cleave *rpoB* for species distinction (15); however, these RFLP patterns are complicated and may result in multiple patterns from a single species. The novel restrictive enzymes system allowed the RFLP patterns in our method to show clear and easy identification, with distinct patterns for *C. canimorsus* and *C. cynodegmi*. The discovery of new nucleotide polymorphisms in the 16S rRNA gene of some *C. cynodegmi* strains hinders species differentiation primer design for conventional PCR. After retrieving the published 16S rRNA sequences of *C. cynodegmi*, we found nucleotide polymorphisms at the same position in the previously published *C. cynodegmi* strains Ccy19 (accession no. GQ167551.1), 11019B-1 (AB851839.1), and LUMC-HA2 (EU124419.1) with occurrence frequencies of 33% (1/3), 14% (1/7), and 17% (1/6), respectively, among the *C. cynodegmi* isolates in the respective studies (13–15). This frequency is similar to our finding of 23% (11/47). These strains were identified using culture-based methods which did not include species-specific PCR identification; therefore, the identification error could not be determined. Nucleotide polymorphism *C. cynodegmi* strains (NP-Ccy) have been observed in Taiwan and other countries, including Switzerland, Japan, and the Netherlands (13–15). This worldwide distribution of NP-Ccy strains affects the accuracy of species-specific PCR surveys, resulting in an overestimated frequency of *C. canimorsus* from the canine oral cavity. If the same primer sets were used in this study, the prevalence of *C. canimorsus* would have been overestimated by 12.2%, instead of 1%, as reported here. The proposed PCR-RFLP method overcomes the misidentification of *C. cynodegmi* as *C. canimorsus* and can be applied in the future for PCR- or culture-based epidemiological surveys of *Capnocytophaga* in small animals. In short, this novel PCR-RFLP method has superior accuracy, efficiency, and ease of application compared to

**FIG 3** Legend (Continued)

pattern type 2 (478 and 987 bp); Ccy2, *C. cynodegmi* belonging to RFLP pattern type 3 (51, 478, and 936 bp); Cca, *C. canimorsus* belonging to RFLP pattern type 1 (14, 51, 286, and 1,114 bp); *C.* sp.: strains 62067N and 91462, the suspected new *Capnocytophaga* species, belonging to RFLP pattern type 4 (51 and 1,414 bp); P, undigested PCR-amplified 16S rRNA product.

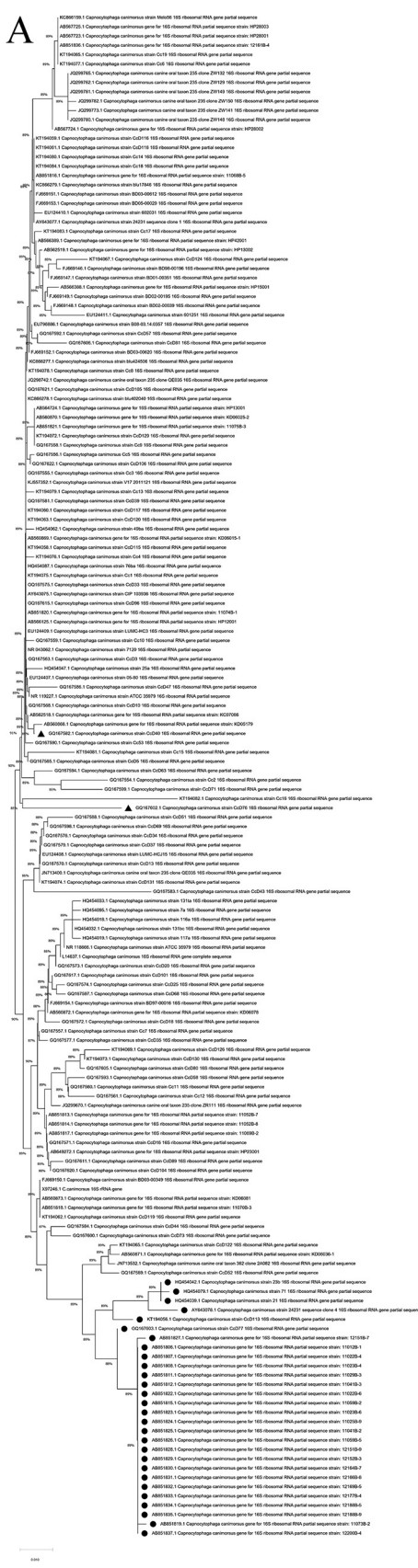

**FIG 4** (Continued)

B

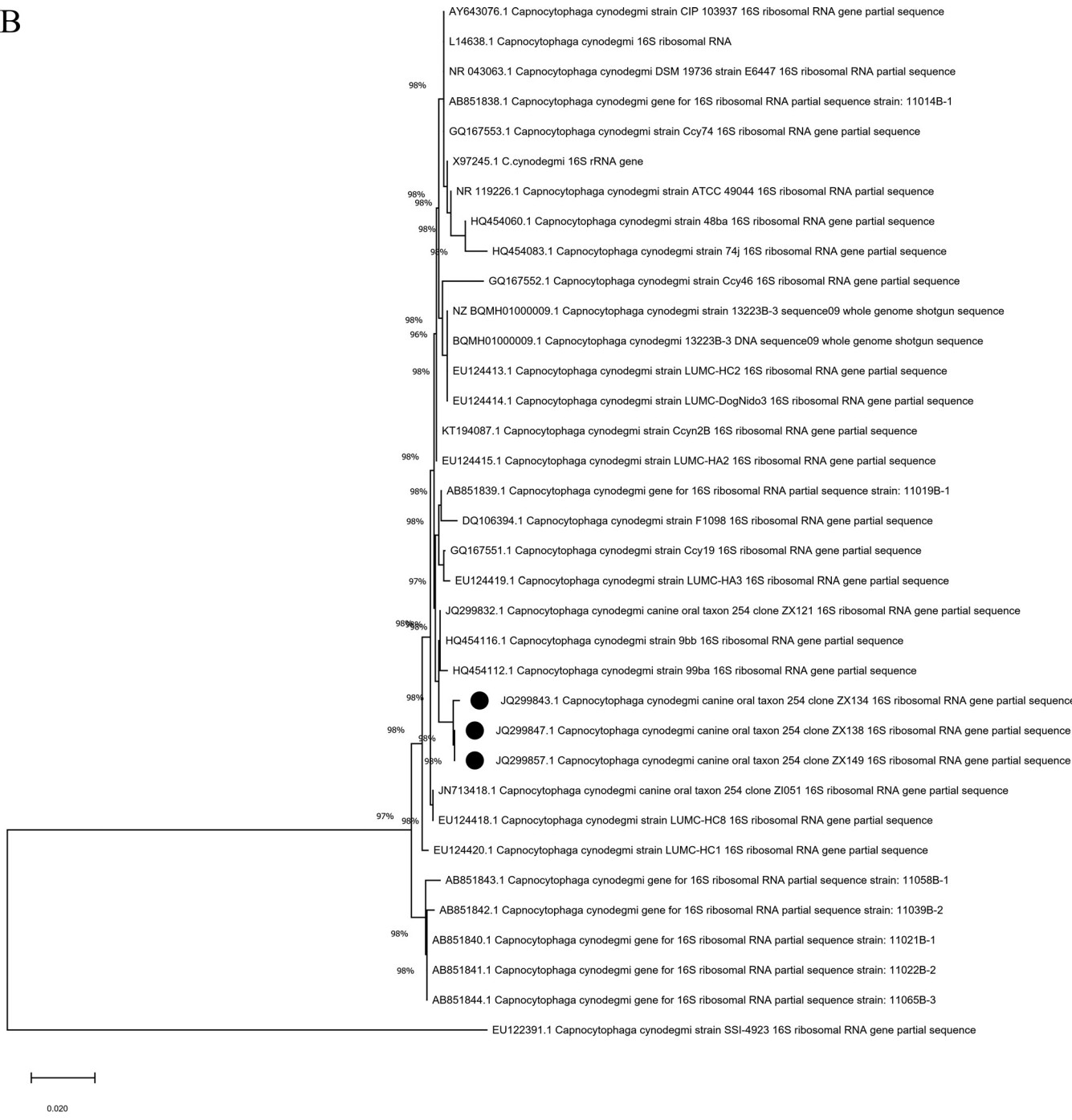

**FIG 4** Phylogenetic trees were constructed based on approximately 440 bp of the 16S rRNA sequence of published *C. canimorsus* and *C. cynodegmi* strains. (A) Published *C. canimorsus* strains. Strains without the expected restriction enzyme (RE) sequences were clustered in a separate clade, except for two strains (indicated by filled triangles). (B) Published *C. cynodegmi* strains. Strains without the expected RE sequences were clustered in a clade that was in close proximity to other clades. Filled dots indicate strains without the expected RE sequence.

other methods. However, this technique can possibly misidentify non-*Capnocytophaga* bacteria with the same/similar RFLP patterns. Genus-specific PCR could thus be used for confirmation, which will be discussed later.

Applying this novel PCR-RFLP analysis to culture-based studies can significantly improve the diversity of *Capnocytophaga* species detected in oral specimens. Moreover, this method is more efficient than direct 16S rRNA gene sequencing for detecting the presence of *C. canimorsus* (PpuMI-specific), *C. cynodegmi* (KpnI-specific), and other species (mismatched patterns of both REs). Considering that non-*Capnocytophaga* bacteria exhibit

**TABLE 2** Molecular techniques for the detection of *C. canimorsus*- and *C. cynodegmi*-related species[a]

| No | Technique | Species | Target gene | Reference |
|---|---|---|---|---|
| 1 | Conventional PCR | *C. canimorsus* and *C. cynodegmi* | 16S rRNA | 3, 13 |
| | | | *rpoB* | 15 |
| 2 | Genus- or species-specific PCR | *C. canimorsus* and *C. cynodegmi* | 16S rRNA (genus- and species-specific) | 5 |
| | | | *rpoB* (species-specific) | 15 |
| 3 | PCR-RFLP | *C. canimorsus* and *C. cynodegmi* | *rpoB* digested with StyI and MseI | 15 |
| | | | 16S rRNA digested with CfoI | 16 |
| | | | 16S rRNA digested with KpnI and PpuMI | This study |
| 4 | PFGE | *C. canimorsus* | Genomic DNA digested with SmaI | 14 |
| 5 | MLST | *C. canis* and *C. canimorsus* | *dnaJ*, *fumC*, *glyA*, *gyrB*, *murG*, *trpB*, and *trf* | 17 |

[a]RFLP, restriction fragment length polymorphism; PFGE, pulsed-field gel electrophoresis; MLST, multilocus sequence typing.

similar morphology and may possess the same RE restriction sites as *Capnocytophaga*, DNA extracted from previous restriction fragments (containing primer pair recognition sites) as genus-specific PCR templates would rule out possible misidentification. Moreover, the method can also be used in certain resource-limited research or clinical units to identify *Capnocytophaga*, especially to detect *C. canimorsus*. For example, genomic DNA was extracted from oral or blood specimens as a template for *C. canimorsus*- and *C. cynodegmi*-specific PCR, followed by the cleavage of positive PCR products with PpuMI. The results showed that *C. canimorsus*-specific PCR cleaved products with a fragment pattern of 15, 198, and 216 bp were identified as *C. canimorsus* (confirmed by our *C. canimorsus* isolate, data not shown), whereas cleaved products 427 bp in size were identified as *C. cynodegmi*. This method facilitates the rapid diagnosis of patients with a history of small animal contact suspected of *C. canimorsus* infection.

Differentiating between *Capnocytophaga* species is challenging because *C. stomatis* cannot be distinguished from other *C. cynodegmi* strains. This finding was consistent with a previous study reporting that *C. stomatis* (isolated from a human patient) and *C. cynodegmi*, which show high genetic similarity, can only be distinguished by whole-genome sequencing (6). This implies that specimens identified as *C. cynodegmi* might also include some strains of *C. stomatis* (even at a low frequency) because these two species cannot be distinguished by 16S rRNA gene sequencing.

Verification of our designed method revealed a high prediction accuracy for the detection of currently published *C. canimorsus* and *C. cynodegmi* strains. There are three possible reasons for profiling failure to identify *C. canimorsus* and *C. cynodegmi* strains: (i) some *C. canimorsus* strains were probably other *Capnocytophaga* species, (ii) incomplete signals were obtained during sequencing, and (iii) sequences between 16S rRNA gene copies were divergent. In the phylogenetic trees of *C. canimorsus* isolated in the respective studies, strains of *C. canimorsus* without the expected RE locus formed a separate clade from that of the other strains (3, 13, 14). Umeda et al. (14) proposed that the *C. canimorsus* strains in a separate clade (21 isolates in that study) were likely to be other *Capnocytophaga* species or subspecies of *C. canimorsus*. After our phylogenetic analysis, we obtained consistent results and considered that these strains were very likely to be other *Capnocytophaga* species. In addition, those strains were highly homogeneous with the others, with only a single deletion at the expected RE position; thus, we speculated that there might have been signal recognition errors in the sequencing process of these strains. Finally, multiple copies of the 16S rRNA gene of *C. canimorsus* strain 24231 have been identified from sequence accession numbers AY643078.1 and AY603477.1, which originated from different clones that are highly similar to the type strains of *C. cynodegmi* and *C. canimorsus*, respectively. The final identification may depend on other genes or whole-genome sequencing. If these reasons for suspicion are all correct, our method can serve as an efficient molecular tool for epidemiological studies in small animals and for diagnosing *C. canimorsus* and *C. cynodegmi* infection in humans.

Identifying *Capnocytophaga* through biochemical testing is difficult due to its fastidious characteristics. The Vitek 2 automatic biochemical identification system (bioMérieux, Marcy-l'Étoile, France) has been used to identify *C. canimorsus* and *C. cynodegmi* in blood and

wound isolates (18), with a 50% rate of correct identification at the genus level (10/20) (18). The Biolog system had low species identification power but a high reference value. Compared to Vitek 2, Biolog is a semiautomatic identification system and requires more manual manipulation to enable a favorable biochemical response by bacteria. Carbohydrate utilization (fructose, sucrose, melibiose, inulin, and raffinose) have been previously used to differentiate between *C. canimorsus* and *C. cynodegmi* (6). In this study, *C. cynodegmi* isolates demonstrated variable carbohydrate utilization. Thus, carbohydrate utilization may not be sufficient for differentiating between *C. canimorsus* and *C. cynodegmi*. Positive oxidase and catalase tests were previously considered to be indicative of *C. canimorsus* and *C. cynodegmi*. However, this study revealed variable oxidase test results in *C. cynodegmi* isolates. Suzuki et al. (17) conducted an MLST analysis and found that the virulence of *C. canis* strains in humans is associated with oxidase activity. Thus, whether oxidase activity affects the virulence of *C. cynodegmi* requires further research for validation.

The prevalence of *Capnocytophaga* differs across countries and is dictated by the detection methods used, such as culture- or PCR-based methods. In general, the frequency of *Capnocytophaga* spp. detected in the canine oral cavity using culture-based methods is lower than that obtained using PCR-based methods (3, 5, 13, 15, 19, 20). For example, the occurrence rate of *C. canimorsus* determined using PCR-based methods ranges from 41% to 74% whereas that with culture-based methods ranges from 5% to 58%. At the species level, the prevalence rate of *C. canimorsus* in Taiwan was lower than that reported in the United States (21.7%) (3), Britain (24%) (20), and Switzerland (58%) (13). All of these countries are at relatively higher latitudes than Taiwan. Thus, climatic factors may also affect the distribution of *Capnocytophaga* species. Cold temperatures are likely to be more suitable for adapting *C. canimorsus* to the canine oral cavity (21). Further, *C. canimorsus* has been shown to adapt to the canine oral cavity by foraging glycans from salivary mucin and N-linked glycoproteins. Increased mucin production can be induced under cold conditions through activation of the transient receptor potential channel melastatin 8 in oral epithelial cells and the tongue (22). Taiwan is located in a subtropical region, and the average annual temperature during specimen sampling was 23.7°C. This relatively high temperature may not be suitable for the presence of *C. canimorsus*, as implied in a study conducted in Brazil wherein the detection rates of *C. canimorsus* and *C. cynodegmi* in dogs with periodontal disease using a PCR-based method were 19% and 66.9%, respectively (23). Additionally, the prevalence of *C. canimorsus* and *C. cynodegmi* in dogs in Shiraz (Iran) was reported as 16% and 28.8%, respectively (24). The average annual temperature in both Brazil and Shiraz is approximately 25°C, similar to that in Taiwan. Thus, factors affecting the adaptation of *Capnocytophaga* spp. to the canine oral cavity need to be investigated further to better understand the epidemiology of *Capnocytophaga*.

Overall, nucleotide polymorphisms in some *C. cynodegmi* strains are reported to impact the accuracy of species-specific PCR methods worldwide. Our results indicate that the newly developed 16S rRNA PCR-RFLP method is a more accurate approach for distinguishing between *C. canimorsus* and *C. cynodegmi*. After verification, our method demonstrated high accuracy, especially for human *C. canimorsus* isolates. Combining the developed genus- and species-specific PCR with our RFLP system provides an improved molecular strategy for the epidemiological investigation of *Capnocytophaga* in small animals and serves as a diagnostic tool for human infections caused by animal bites.

## MATERIALS AND METHODS

**Specimen collection, culture, and isolation of *Capnocytophaga*.** Specimens were collected from the oral cavity of dogs brought to the Veterinary Medicine Teaching Hospital (VMTH), National Chung Hsing University (NCHU), or to a regional animal shelter from January to September 2020. Oral specimens were obtained with sterile cotton swabs by firmly rubbing the gingival margin and tongue of each dog in a conscious state. Basic information, including age, sex, breed, recent history of antibiotic use, main diet, and stage of periodontal disease was recorded. This study was approved by the Institutional Animal Care and Use Committee (approval no. 109-064). All owners provided informed consent.

Each swab was immediately plated onto a selective medium, prepared using Columbia agar (Becton, Dickinson, and Company, Franklin Lakes, NJ, USA) with 10% sheep's blood (Taiwan Prepared Medium, Taipei, Taiwan), a polyvitaminic supplement (Supplement VX; Becton, Dickinson and Company, Franklin Lakes, NJ, USA), and an antibiotic mixture containing 2.5 $\mu$g/mL trimethoprim (Sigma-Aldrich, St. Louis, MO, USA), 5 $\mu$g/

mL vancomycin (Sigma-Aldrich), 3.75 $\mu$g/mL colistin (Sigma-Aldrich), and 15 U/mL polymyxin B (Sigma-Aldrich). The plates were incubated at 37°C in sealed candle jars for 2 to 3 days. Colonies showing the morphological characteristics of *Capnocytophaga* (1 to 2 mm in size, smooth, white to off-white in color, and circular) were examined by Gram staining to confirm their microscopic morphology under a light microscope. Colonies presenting both colony and bacterial cell morphological characteristics that matched *Capnocytophaga* were subcultured in fresh medium. A maximum of 20 matched colonies per plate was selected for further analysis.

**Identification and phylogenetic analysis of *Capnocytophaga* isolates.** Isolated and purified colonies (see previous section) subsequently underwent genus-specific and/or species-specific PCR to confirm that the selected colonies were *Capnocytophaga* sp. The primers and PCR conditions used were previously described by Suzuki et al. (5). Colonies with positive PCR results were propagated for further DNA extraction and the enriched bacterial cells from the subcultured plates were stocked in cryogenic vials with preservation beads (Taiwan Prepared Media, Taipei, Taiwan) and stored at −80°C. DNA extraction was performed using an AxyPrep bacterial genomic DNA miniprep kit (Axygen Scientific Inc., Union City, CA, USA) according to the manufacturer's instructions. The 16S rRNA gene was subsequently amplified from extracted DNA and sequenced for species identification. To validate isolates not recognized by genus-specific or species-specific PCR and for colonies that exhibited the morphological features of *Capnocytophaga* but had negative PCR results, confirmation was performed using 16S rRNA sequences.

Amplification of the 16S rRNA gene from previously extracted DNA was performed using primers 27F (5′-AGAGTTTGATCCTGGCTCAG-3′) and 1492R (5′-GGTTACCTTGTTACGACTT-3′) (25). The thermal protocol was as follows: initial denaturation at 94°C for 2 min; 30 cycles of denaturation at 94°C for 15 s, annealing at 55°C for 15 s, and extension at 72°C for 90 s; and final extension at 72°C for 10 min in a MiniAmp Plus Thermal Cycler (Thermo Fisher Scientific, Waltham, MA, USA). Approximately 1,465 bp of the amplified products was analyzed by electrophoresis on a 1% agarose gel. The PCR products were pretreated using ExoSAP-IT PCR Product Cleanup Reagent (Thermo Fisher Scientific) and sequenced using an ABI 3730 automated DNA sequencer (Applied Biosystems, Rockville, MD, USA). Sequences were compiled using SeqMan software (version 7.0; DNASTAR, Inc., Madison, WI, USA) and submitted to the GenBank database. The accession numbers of the 16S rRNA sequences of the cultured *Capnocytophaga* species isolated from the canine oral cavity are MZ314747 to MZ314793, MZ314818, and MZ314826 to MZ314827. The reference strains retrieved from GenBank included three strains of *C. canimorsus* (ATCC 35979, CP022382, and NC015846), three strains of *C. cynodegmi* (ATCC 49044, AY643076, and CP022378), one strain of *C. felis* (accession no. LC411961), one strain of *C. canis* (accession no. LC100036.1), and one strain of *C. stomatis* (CP022387).

The 16S rRNA sequences were aligned using ClustalW in MEGA-X software (26). A phylogenetic tree was constructed using the neighbor-joining method, and genetic distances were calculated using the Kimura 2-parameter model with 1,000 bootstrap replications. Phylogenetic analysis and tree construction were conducted using MEGA-X software.

**Biochemical tests.** Biochemical tests were performed using a Biolog GEN III MicroPlate test panel (Biolog Inc., Hayward, CA, USA) according to the manufacturer's instructions. Briefly, bacteria were enriched on blood agar plates and homogenously mixed with IF-C inoculation fluid until the turbidity reached 65%. The inoculate was added to the microplate and cultured at 37°C in a candle jar. Color-change analyses and interpretation of the results were conducted using Biolog Microbial Identification Systems software (Biolog Inc.) at 24 and 48 h after incubation. The system compared the test strain results against an established database and used similarity indices to determine the species ID; otherwise, the genus ID or no ID was shown.

Oxidase and catalase tests were performed on each isolated *Capnocytophaga* strain. The catalase tests were performed using 3% hydrogen peroxide solution (Union Chemical, Hsinchu, Taiwan) and the oxidase tests were performed using oxidase reagent (Becton, Dickinson and Company).

**16S rRNA PCR-RFLP.** The aforementioned 16S rRNA gene sequence alignment of the isolates and reference strains indicated differences between *C. canimorsus* and *C. cynodegmi*. The 16S rRNA gene sequences of *C. canimorsus* and *C. cynodegmi* strains with high concordance in their variance sites were identified. We then used SeqMan software (version 7.0; DNASTAR, Inc.) to search for restriction enzymes that could differentiate among the divergent sequences. We selected an RFLP system with the most distinctive restrictive patterns and validated it *in silico*. The REs KpnI and PpuMI were finally chosen for the RFLP system. The RE reaction mixture was prepared according to the manufacturer's instructions and contained 1 $\mu$L amplified PCR product, 2 $\mu$L Green Buffer, 1 $\mu$L KpnI, and 1 $\mu$L PpuMI (Thermo Fisher Scientific). Digestion was performed at 37°C for 10 min followed by inactivation at 80°C for 5 min. Restriction fragments were analyzed by electrophoresis on 1.5% agarose gels.

***In silico* verification of predicted RE sites in published *C. canimorsus* and *C. cynodegmi* strains.** We retrieved all published 16S rRNA gene sequences of *C. canimorsus* and *C. cynodegmi* from GenBank and the RefSeq data bank of NCBI (27). The sequence search filter used the default settings and sequence length was limited from 200 to 1,600 bp. The results classified by taxon showed 275 matched sequences of *C. canimorsus* and 84 matched sequences of *C. cynodegmi*. After phylogenetic analysis, whole-genome shotgun sequences, sequences shorter in length than the RE position, and other sequences that were too diverse from the 16S rRNA gene were removed. A phylogenetic tree was subsequently constructed based on comparable 16S rRNA sequences of the published *C. canimorsus* and *C. cynodegmi* strains. The method used for phylogenetic tree construction was the same as that described in the previous section "Identification and phylogenetic analysis of *Capnocytophaga* isolates."

**Statistical analysis.** The association between individual factors (including sex, age, weight, main diet, tooth-brushing habits, antibiotic history, periodontal disease stage, and whether the dogs were owned or sheltered) and the presence of *Capnocytophaga* spp. was analyzed using Fisher's exact test. Differences were

considered significant at $P < 0.05$. Data analysis was performed using the Statistical Analysis System version 9.4 (SAS Institute Inc., Cary, NC, USA).

**Data availability.** The authors confirm that the data supporting the findings of this study are available within the article and its supplemental material.

## SUPPLEMENTAL MATERIAL

Supplemental material is available online only.

**SUPPLEMENTAL FILE 1**, XLSX file, 0.01 MB.
**SUPPLEMENTAL FILE 2**, PDF file, 0.01 MB.
**SUPPLEMENTAL FILE 3**, XLSX file, 0.03 MB.
**SUPPLEMENTAL FILE 4**, XLSX file, 0.01 MB.

## ACKNOWLEDGMENTS

We thank Po-Yu Liu (Taichung Veterans General Hospital, Taiwan) for providing constructive suggestions regarding article presentation and knowledge about human patients infected by *Capnocytophaga* species. We acknowledge Meng-Wei Ko (Post-doctoral Scholar, School of Medicine, University of California, San Diego) for editing the manuscript.

We declare that we have no conflicts of interest related to this study.

This research did not receive any specific grants from funding agencies in the public, commercial, or not-for-profit sectors.

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
