## [Reviewer comments · Microbiology Spectrum]

Microbiology Spectrum

A Novel 16S rRNA PCR-Restriction Fragment Length Polymorphism Assay to Accurately Distinguish Zoonotic *Capnocytophaga canimorsus* and *Capnocytophaga cynodegmi*

Cheng-Hung LAI, Yu-Sin LIN, Chao-Min Wang, Poa-Chun CHANG, and Wei-Yau SHIA

Corresponding Author(s): Wei-Yau SHIA, National Chung Hsing University

Review Timeline:

Submission Date:	July 28, 2022
Editorial Decision:	November 19, 2022
Revision Received:	December 15, 2022
Editorial Decision:	February 9, 2023
Revision Received:	February 23, 2023
Accepted:	May 4, 2023

Editor: John Osei Sekyere

Reviewer(s): Disclosure of reviewer identity is with reference to reviewer comments included in decision letter(s). The following individuals involved in review of your submission have agreed to reveal their identity: TINTU ABRAHAM (Reviewer #1); Özlem Şahan Yapıcıer (Reviewer #2)

Transaction Report:

DOI: <https://doi.org/10.1128/spectrum.02916-22>

November 17, 2022

Dr. Wei-Yau SHIA
National Chung Hsing University
Veterinary Medicine
145, XingDa Rd.
Taichung 402
Taiwan

Re: Spectrum02916-22 (A Novel 16S rRNA PCR-Restriction Fragment Length Polymorphism Assay to Accurately Distinguish Zoonotic *Capnocytophaga canimorsus* and *Capnocytophaga cynodegmi*)

Dear Dr. Wei-Yau SHIA:

Please ensure to address all the concerns of the reviewers to maximum satisfaction before your manuscript can be considered or accepted for publication.

Link Not Available

Sincerely,

John Osei Sekyere

Journals Department
Reviewer comments:

Reviewer #1 (Comments for the Author):

NA

Reviewer #2 (Comments for the Author):

Dear author/ authors,

Investigation of the novel molecular assay to distinguish *Capnocytophaga canimorsus* and *C. cynodegmi* were aimed in the present study.

1. The scientific set up and methods of the study is convenient. Nevertheless, the conventional isolation and identification methods could be explained more fluent.

3. In the study, also, there are some deficiencies in terms of microbiological terminology.

4. The authors can find the revision recommendations defined on the manuscript uploaded to the manuscript central.

Sincerely,

Reviewer #4 (Comments for the Author):

Spectrum02916-22

The manuscript title "A Novel 16S rRNA PCR-Restriction Fragment Length Polymorphism Assay to Accurately Distinguish Zoonotic *Capnocytophaga canimorsus* and *C. cynodegmi*" is about the new molecular tool for the epidemiology study of *Capnocytophaga* in small animals and for the rapid diagnosis of human *C. canimorsus* infections. The manuscript is good written but needs improve in the experimental studies and lack of innovative points. I have some comments to take in consideration to the authors:

1. The main problem of this work is the lack of the innovative technique.

2. If the study is about detection of the *Capnocytophaga canimorsus* and *C. cynodegmi*. Why the authors analyzed the Virulence factor analysis?

3. There are more studies published about molecular techniques about the detection of these pathogens? Include table about it.

4. Add advantages and disadvantages of using of this technique.

Reviewer #5 (Comments for the Author):

In this manuscript, Lai et al. developed a novel 16S rRNA PCR-RFLP method to accurately distinguish between *C. canimorsus* and *C. cynodegmi*. This is an interesting approach to differentiate related species. I hope the authors view these comments as being constructive and designed to help strengthen their manuscript.

Major Comments:

Figure 1: This is a challenging figure to interpret. Figure 1A is a phylogenetic tree based on 16S rRNA gene sequences. Figure 1B uses the same phylogenetic tree, based on 16S rRNA gene sequences, and adds the molecular and phenotypic testing results in color-coded boxes. How should these two trees be interpreted differently? How do A-E type strains relate to 1-4 type strains? The text describes how types A, C, D, and E strains are consistent with PCR and 16S rRNA gene sequence analysis. The RFLP type strains are described later in the manuscript, but it is difficult to understand how the data should be interpreted as presented in figure 1 (e.g., all A and E type strain are consistent with type 3).

Minor Comments:

Lines 54-55: *C. canimorsus* usually causes systemic infections in immunocompromised individuals or compared to other *Capnocytophaga* species?

Line 59: What is meant by "potential hazards"? Is this referring to virulence factors of zoonotic *Capnocytophaga* in the canine oral cavity?

Lines 137-138: This is not a full sentence.

Line 171: The presence of *Capnocytophaga* may represent infection or colonization as the organism is part of the normal oral microbiota.

Lines 190-193: From a clinical microbiology perspective, most strains of *C. canimorsus* and *cynodegmi* are predicted to be oxidase and catalase positive in phenotypic identification testing. The significance of these virulence factors, or the lack of the virulence factors for strain 62067N, is not discussed any further.

Line 302: Differentiating, not differencing.

Line 465: Please amend figure legend to reflect that Cca should demonstrate the type 1 pattern (14, 51, 286, and 1114 bp) and that only the 286 and 1114 bp fragments are shown.

Staff Comments:

Preparing Revision Guidelines

Please return the manuscript within 60 days; if you cannot complete the modification within this time period, please contact me. If you do not wish to modify the manuscript and prefer to submit it to another journal, please notify me of your decision immediately so that the manuscript may be formally withdrawn from consideration by Microbiology Spectrum.

**A Novel 16S rRNA PCR-Restriction Fragment Length Polymorphism Assay to**
**Accurately Distinguish Zoonotic *Capnocytophaga canimorsus* and *C. cynodegmi***

Cheng-Hung LAI ^{1,2}, Yu-Sin LIN ¹, Chao-Min WANG ³, Poa-Chun CHANG ⁴, Wei-Yau SHIA ^{1,2,*}

¹Department of Veterinary Medicine, College of Veterinary Medicine, National Chung Hsing
University, Taichung, Taiwan

²Veterinary Medical Teaching Hospital, College of Veterinary Medicine, National Chung Hsing
University, Taichung, Taiwan

³Department of Veterinary Medicine, National Chiayi University, 580 Xinmin Rd., Chiayi City,
Taiwan.

⁴Graduate Institute of Microbiology and Public Health, National Chung Hsing University, Taichung,
Taiwan

***Corresponding author**

Wei-Yau Shia

Department of Veterinary Medicine, College of Veterinary Medicine, National Chung Hsing
University, 145 Xingda Road, Taichung 402, Taiwan

Tel: +886-4-22840368

Fax: +886-4-22852016

Email: vmwyshia@nchu.edu.tw

**Abstract**

The zoonotic bacteria *Capnocytophaga canimorsus* and *C. cynodegmi*, the predominant
*Capnocytophaga* species in the canine oral biota, are reported to cause human local wound
infections or lethal sepsis mostly transmitted through dog bites. Molecular surveying of these two
*Capnocytophaga* species using conventional 16S rRNA-based polymerase chain reaction (PCR) is
not always accurate due to high genetic homogeneity. This study isolated *Capnocytophaga* spp.
from canine oral cavity, subsequent identified them using 16S rRNA and phylogenetic analysis. A
novel 16S rRNA PCR-restriction fragment length polymorphism (RFLP) method was designed
from our isolates and validated using published *C. canimorsus* and *C. cynodegmi* 16S rRNA
sequences. The result showed that 51% of dogs carried *Capnocytophaga* spp. Among which, *C.*
*cynodegmi* (47/98, 48%) was the predominant isolated species along with one strain of *C.*
*canimorsus* (1/98, 1%). Alignment analysis of 16S rRNA sequences revealed specific site
nucleotide diversity in 23% (11/47) of the *C. cynodegmi* isolates, which were misidentified as *C.*
*canimorsus* using previous reported species-specific PCR. Four RFLP types could be classified
from all isolated *Capnocytophaga* strains. The method demonstrated superior resolution in
distinguishing *C. cynodegmi* (with site-specific polymorphism) from *C. canimorsus*, especially in
distinguishing *C. canimorsus* from other *Capnocytophaga* species. After validation *in silico*, this
method revealed overall detection accuracy of 84%, notably, accuracy reached 100% in *C.*
*canimorsus* strains isolated from human patients. The method reported here is a useful molecular
tool for the epidemiology study of *Capnocytophaga* in small animals and for the rapid diagnosis of
human *C. canimorsus* infections.

**Keywords:** *Capnocytophaga*; phylogenetic analysis; dogs; PCR-restriction fragment length
polymorphism (PCR-RFLP); zoonosis

**1. Introduction**

Gram

The genus *Capnocytophaga* comprises fastidious ~~gram~~-negative, thin, or filamentous rods with
tapered or spindle-shaped ends, and facultative anaerobic and capnophilic bacteria [1, 2]. Four
*Capnocytophaga* species (~~spp.~~) have been isolated from the canine oral cavity; among these, *C.*
*canimorsus* and *C. cynodegmi* are more predominant than *C. canis* and *C. stomatis* [3-6]. The most
common transmission pathway is *via* dog bites, followed by scratches or close contact [7]. Although
infected patients may present varying symptoms ranging from mild to fulminant,
immunocompromised patients tend to show more severe clinical signs [7, 8]. *C. canimorsus* usually
causes systemic infections, including cellulitis, meningitis, and sepsis [9]. *C. cynodegmi* is
considered less pathogenic than *C. canimorsus*, since most *C. cynodegmi* infected cases develop
local wound infection (cellulitis) and rarely advance to systemic infection [10, 11].

In some epidemiological studies, culture- and/or direct polymerase chain reaction (PCR)-based
techniques are often used to evaluate the potential hazards of zoonotic *Capnocytophaga* spp. in the
canine oral cavity. Among the species investigated, *C. canimorsus* has been the most extensively
surveyed [3, 5, 12-15]. PCR-based methods usually reveal a higher prevalence of *Capnocytophaga*
compared to culture-based methods, owing to the fastidious property of this genus and the high
sensitivity of PCR. However, without the culture process, the phenotypic and other important
genotypic characteristics of *Capnocytophaga* cannot be determined.

Genetic identification of *Capnocytophaga* spp. is commonly performed by 16S rRNA
sequencing [3, 15]. However, considering the high homogeneity of the 16S rRNA gene between *C.*
*canimorsus* and *C. cynodegmi*, other techniques including pulsed-field gel electrophoresis,
**MLST**
multi-locus sequence analysis, and PCR-restriction fragment length polymorphism (RFLP), have
also been applied to discriminate between canine *Capnocytophaga* spp. [14-17]. Suzuki et al.
developed genus- and species-specific PCR targeting the 16S rRNA gene to allow rapid
identification and distinction between *C. canimorsus* and *C. cynodegmi* strains isolated from dogs
and cats [5]. These primer sets have also been widely adopted in other studies to investigate the

73 prevalence of *Capnocytophaga* spp. in small animals [4, 14, 18].

In this study, during the culture-based epidemiological investigation of *Capnocytophaga* in
dogs, we discovered that the 16S rRNA sequence polymorphism at the specific location of *C.*
*cynodegmi* strains showed ambiguous results on the previously established species-specific PCR.
Therefore, we utilized the isolated strains to develop a novel 16S rRNA PCR-RFLP method which
accurately distinguished between *C. canimorsus* and *C. cynodegmi*, and demonstrated its potential
application in future epidemiological studies and disease diagnosis.

**2. Materials and Methods**

*2.1. Specimen collection, culture, and isolation of Capnocytophaga*

Specimens were collected from the oral cavity of dogs brought to the Veterinary Medicine
Teaching Hospital, National Chung-Hsing University (VMTH, NCHU), or to a regional animal
shelter from January 2020 to September 2020. Oral specimens were obtained by firmly rubbing the
gingival margin and tongue with sterile cotton swabs in each dog in a conscious state. Basic
information including age, sex, breed, recent history of antibiotic use, main diet, and stage of
periodontal disease was recorded. This study was approved by the Institutional Animal Care and
Use Committee (IACUC) (approval number:109-064). All owners provided informed consent.

Each swab was immediately plated onto a selective medium, prepared using Columbia agar
(Becton, Dickinson, and Company, Franklin Lakes, NJ, USA) with 10% sheep's blood (Taiwan
Prepared Medium, Taipei, Taiwan), polyvitaminic supplement (Supplement VX, Becton, Dickinson,
**, Brand country?** and Company), and the antibiotic mixture containing 2.5 µg/ml trimethoprim (Sigma–Aldrich, St.
Louis, MO, USA), 5 µg/ml vancomycin (Sigma–Aldrich), 3.75 µg/ml colistin (Sigma–Aldrich), and
15 U/ml polymyxin B (Sigma–Aldrich). The plates were incubated at 37 °C in sealed candle jars for
2–3 days. Colonies showing morphological characteristics of *Capnocytophaga* (1–2 mm in size,
smooth, white to off-white in color, and circular) were examined by Gram staining to confirm
~~bacterial~~ morphology under a light microscope. Colonies presenting both colony and bacterial cell
microscopic

morphological characteristics that matched *Capnocytophaga* were sub-cultured in fresh medium. A
maximum of 20 matched colonies per plate was selected for further analysis.

2.2. Identification of *Capnocytophaga* colonies

bacterial identification?

Genus-specific PCR and/or species-specific PCR were used for ~~colony screening~~. The primers
and PCR conditions used were previously described by Suzuki et al. [5]. Colonies with positive
PCR results were propagated for further DNA extraction, and bacterial cell stocks were stored at -80
105 °C. The extracted DNA was subsequently amplified from 16S rRNA genes and sequenced for
species identification (described in Section 2.5). DNA extraction and 16S rRNA identification were
still performed for colonies that demonstrated the morphological characteristics of *Capnocytophaga*
but had negative genus- or species-specific PCR results.

2.3. Virulence factor analysis

Biochemical tests (do we use just catalase and oxidase or any other biochemical tests?)

In this study, oxidase and catalase tests were performed on each isolated *Capnocytophaga* strain.
A catalase test was performed using 3% hydrogen peroxide solution (Union Chemical, Hsinchu,
Taiwan), and oxidase tests were performed using oxidase reagent (Becton, Dickinson and
Company).
Country

2.4. Phylogenetic analysis of 16S rRNA sequences

DNA was extracted from all strains using the AxyPrep bacterial genomic DNA miniprep kit
(Axygen Scientific Inc., Union City, CA, USA) according to the manufacturer's instructions.
Amplification of the 16S rRNA gene was performed using primers 27F
(5'-AGAGTTTGATCCTGGCTCAG-3') and 1492R (5'-GGTTACCTTGTTACGACTT-3') [19]. The
thermal protocol was as follows: initial denaturation at 94 °C for 2 min; 30 cycles of denaturation at
94 °C for 15 s, annealing at 55 °C for 15 s, and extension at 72 °C for 90 s; and final extension at 72
121 °C for 10 min in a Miniamp Plus Thermal Cycler (Thermo Fisher Scientific, Waltham, MA, USA).
Approximately 1465 bp of the amplified products were analyzed by electrophoresis on a 1%
agarose gel and sequenced using an ABI 3730 automated DNA sequencer (Applied Biosystems).
Sequences were compiled using SeqMan software (SeqMan[®], Version 7.0, DNASTAR, Madison,

WI, USA) and submitted to the GenBank database. The accession numbers of 16S rRNA sequences
of the cultured *Capnocytophaga* species isolated from the canine oral cavity are
MZ314747-MZ314793, MZ314818, and MZ314826-MZ314827. The reference strains retrieved
from GenBank included three strains of *C. canimorsus* (ATCC35979, CP022382, and NC015846),
three strains of *C. cynodegmi* (ATCC49044, AY643076, and CP022378), one strain of *C. felis*
(accession No. LC411961), one strain of *C. canis* (accession no. LC100036.1), and one strain of *C.*
*stomatis* (CP022387).

16S rRNA sequences were aligned using ClustalW in MEGA-X software [20]. A phylogenetic
tree was constructed using the neighbor-joining method, and genetic distances were calculated
using the Kimura 2-parameter model with 1000 bootstrap replications. Phylogenetic analysis and
tree construction were conducted using MEGA-X software.

2.5. 16S rRNA PCR-RFLP

According to the aforementioned aligned 16S rRNA gene sequences of the isolated
*Capnocytophaga* strains and reference strains. Comparative alignment of 16S rRNA sequences
revealed differences between *C. canimorsus* and *C. cynodegmi*, SeqMan software (SeqMan[®],
Version 7.0, DNASTAR, Madison, WI, USA) was used to identify the relevant restriction enzyme
sites. The restriction enzyme (RE) reaction mixture was prepared according to the manufacturer's
instructions and contained 1 µl amplified PCR product, 2 µl Green Buffer, 1 µl KpnI, and 1 µl
PpuMI (Thermo Scientific, Waltham, MA, USA). Digestion was performed at 37 °C for 10 min
followed by inactivation at 80 °C for 5 min. Restriction fragments were analyzed by electrophoresis
on 1.5% agarose gels.

2.6. Verification of predicted RE sites in published *C. canimorsus* and *C. cynodegmi* strains in 147 *silico*

We retrieved all published 16S rRNA gene sequences of *C. canimorsus* and *C. cynodegmi* from
GenBank and the RefSeq databank of NCBI [21]. The sequence search filter used the original
settings and sequence length was limited from 200 to 1600 bp. The results classified by taxon

showed 275 matched sequences of *C. canimorsus* and 84 matched sequences of *C. cynodegmi*. After
phylogenetic analysis, whole-genome shotgun sequences, sequence lengths shorter than the RE
position, or other sequences that were too diverse from the 16S rRNA gene were also removed. A
phylogenetic tree was subsequently constructed based on comparable 16S rRNA sequences of the
published *C. canimorsus* and *C. cynodegmi* strains. The method of phylogenetic tree construction
was the same as that described in the previous section, “Phylogenetic analysis of 16S rRNA
sequences”.

2.7. Statistical analysis

The association between individual factors (including sex, age, weight, main diet, tooth
brushing habits, antibiotic history, periodontal disease stage, and whether the dogs were owned or
sheltered) and the presence of *Capnocytophaga* spp. was analyzed using Fisher’s exact test.
Differences were considered significant at p -values < 0.05 . Data analysis was performed using the
Statistical Analysis System version 9.4 (SAS Institute Inc., Cary, NC, USA).

3. Results

3.1. Prevalence, virulence factors, and phylogenetic analysis of 16S rRNA sequences

Oral swabs were collected from 98 dogs (50 males and 48 females), including 82 domestic and
16 shelter dogs. The median age of collected dogs was 8 years (range, 0.25–17 years). Relevant
individual parameters were summarized and none of the basic individual parameters was
significantly correlated with the presence of *Capnocytophaga* (Supplementary Table 1). The
prevalence of *Capnocytophaga* infection in the canine oral cavity was 51% (49/98). Fifty
*Capnocytophaga* strains were isolated from 49 dogs. After sequencing the amplified 16S rRNA of
these 50 strains and trimming the poor signal sequences from both ends, sequences of
approximately 1331 bp were used for the BLAST and subsequent phylogenetic analysis. Combining
the 16S rRNA BLAST and phylogenetic analysis results, *C. cynodegmi* showed the highest
prevalence (47.96%; 47/98) among all sampled dogs. There was also one *C. canimorsus* (1/98,

1.02%) isolate and two strains (62067N and 91462) identified as *C. cynodegmi* with identities of
95.7% and 97.4%, respectively, which were likely to be new species. Two different
*Capnocytophaga* species were present in a single specimen: *C. cynodegmi*, and a suspected new
species (strain 62067N).

A phylogenetic tree was constructed based on the 16S rRNA sequences of the 50 isolates and
nine reference strains obtained from GenBank (Fig. 1). Although the 16S rRNA sequences of 47
strains of *C. cynodegmi* showed high similarity, three clades were identified with 99% bootstrap
support. Strain 91462 belonged to the same clade as *C. canis*, with 99% bootstrap support; however,
the 16S rRNA BLAST results showed 97.4% identity with *C. cynodegmi*, suggesting that further
species identification was needed. Another unidentified *Capnocytophaga* strain (62067N) was
shown to be entirely independent of other canine-related *Capnocytophaga* strains (Fig. 1A). The
reference strain *C. stomatis* H2177 (GenBank accession number NZ_CP022387.1) was not well
distinguished from *C. cynodegmi* according to the 16S rRNA-based phylogenetic tree.

For basic virulence factor assays, all 47 strains of *C. cynodegmi* were catalase positive, and
87.2% (41/47) were oxidase-positive (Fig. 1B). There was one *C. canimorsus* strain showed
positive results for both catalase and oxidase tests. The unidentified *Capnocytophaga* strain
(62067N) was the only strain negative for both catalase and oxidase tests.

3.2. Effect of site-specific sequence polymorphism on genus- and species-specific PCR

Thirteen *Capnocytophaga* isolates are inconsistent in BLAST results for 16S rRNA genes
between the genus- and species-specific PCR. Eleven strains identified as *C. cynodegmi* based on
the 16S rRNA gene, with identities ranging from 98.5–99.7%, were positive for both *C.*
*canimorsus*- and *C. cynodegmi*-specific PCR. Two strains (91462 and 62067N) identified as *C.*
*cynodegmi* with 97.4% and 95.7% identity, respectively, yielded negative results for both genus-
and species-specific PCR. The negative PCR results were very likely due to sequences marked
differently from the recognition site of the shared forward primer CaL2 in genus- and

species-specific PCR (Fig. 2A and 2B). According to sequence alignment with the *C.*
*cynodegmi*-specific PCR reverse primer, the 50 *Capnocytophaga* strains were classified into five
different types (A–E; Fig. 1B and 2C). Strains belonging to types A, C, D, and E are consistent with
species-specific PCR results and 16S rRNA gene sequence BLAST analysis. Type B *C. cynodegmi*
isolates were all positive to *C. canimorsus*-specific PCR (Fig. 1B). The reverse primer binding
sequences of these strains differed from the *C. canimorsus*-specific reverse primer by only two
nucleotides (positions 479 and 483) (Fig. 2C). This explained the mismatched results were due to
the substitution of nucleotide G for A at position 482 in the 16S rRNA gene.

3.3. Differentiation of *C. canimorsus* and *C. cynodegmi* by 16S rRNA PCR-RFLP

Amplified 16S rRNA gene products of 50 *Capnocytophaga* isolates were treated with restriction
enzymes (REs), KpnI and PpuMI, and followed by gel electrophoresis (Fig. 3). Only the PCR
product from *C. canimorsus* was cleaved by PpuMI. In contrast, the PCR products of *C. cynodegmi*
and the two unidentified strains (91462 and 62067N) were cleaved only by KpnI at different sites.
Using the 16S rRNA gene of the reference strain, four patterns were predicted *in silico* by
combining KpnI and PpuMI (Table 1 and Fig. 1B). The type 1 pattern was represented by fragment
patterns of 14, 51, 286, and 1114 bp in *C. canimorsus* reference strains; our isolates showed
consistent results (Fig. 3). The *C. cynodegmi* isolates showed two different restriction fragment
patterns after treatment with KpnI: the type 2 pattern (478 and 987 bp) was found in six isolates,
whereas the type 3 pattern (51, 478, and 936 bp) was found in 41 isolates and three reference strains.
The reference strain *C. stomatis* showed the same pattern as *C. cynodegmi* (type 3), as predicted *in*
*silico* (Table 1). The last pattern, type 4 (51 and 1414 bp), was observed in the reference strains of *C.*
*felis* and *C. canis*, and in our unidentified strains (Table 1). In summary, *C. canimorsus* (fragments
286 bp and 1114 bp) and *C. cynodegmi* (fragments 478 bp and 936 bp or 987 bp) can be clearly and
easily differentiated by gel electrophoresis of the rRNA PCR amplicons digested by KpnI and
PpuMI (Fig. 3).

3.4. Verification of expected REs in the published *C. canimorsus* and *C. cynodegmi* strains

After phylogenetic analysis, 172 *C. canimorsus* and 35 *C. cynodegmi* 16S rRNA gene
sequences were collected. The sequence of the expected RE site of each strain is presented in the
supplementary data (Supplementary Fig. 1 and 2). 83% (142/172) of *C. canimorsus* and 91% (32/35)
of *C. cynodegmi* strains were consistent with the designed 16S rRNA PCR-RFLP results. The
overall accuracy for RE detection was 84%. Importantly, the accuracy of *C. canimorsus* strains
detection isolated from human patients was 100%. All those strains showed inconsistent results (30
*C. canimorsus* and 3 *C. cynodegmi*) without the detection of the predicted restriction sites were
isolated from dogs and cats. Failure to identify the expected RE in *C. canimorsus* strains mainly
(28/30, 93.3%) resulted from the C/T substitution at position 289 and the G/T substitution at
position 300. The other two *C. canimorsus* strains (accession numbers GQ167582.1 and
GQ167602.1) did not show the expected RE was owing to a single deletion at the expected RE
position (position 288). A similar finding was obtained in three *C. cynodegmi* strains, where the
deletion occurred at position 475 (Supplementary Fig. 1 and 2).

The phylogenetic tree indicated that 28 of the 30 *C. canimorsus* strains lacking the expected RE
site were allocated to a separate clade, with 89% bootstrap support (Fig. 4A). The aforementioned 2
strains of *C. canimorsus* with a single deletion at position 288 were phylogenetically close related to
the other *C. canimorsus* strains (Fig. 4A). Similarly, 3 strains of *C. cynodegmi* with single deletion
at the expected RE site were in proximity to the neighboring clades of other *C. cynodegmi* strains
(Fig. 4B).

**4. Discussion**

[revised manuscript text omitted]

Differencing *Capnocytophaga* species is challenging since *C. stomatis* cannot be distinguished
from other *C. cynodegmi* strains. This finding was consistent with a previous study reporting that *C.*
*stomatis* (isolated from a human patient) and *C. cynodegmi*, which show high genetic similarity, can
only be distinguished through whole-genome sequencing [6]. This implies that specimens identified
in the presence of *C. cynodegmi* might also include some strains of *C. stomatis* (even with a low

frequency) as these two species cannot be distinguished by 16S rRNA gene sequencing.

Verification of our designed method revealed high prediction accuracy for the detection of *C.*
*canimorsus* and *C. cynodegmi* strains published to date. There are 3 possible reasons for profiling
failure to identify *C. canimorsus* and *C. cynodegmi* strains: 1) some *C. canimorsus* strains were
probably other *Capnocytophaga* species, 2) incomplete signals during sequencing, and 3)
divergence of sequences between 16S rRNA gene copies. In the phylogenetic trees of *C.*
*canimorsus* isolated in the respective studies, strains of *C. canimorsus* without the expected RE
locus formed a separate clade from the other strains [3, 13, 14]. Umeda *et al.* proposed that *C.*
*canimorsus* strains in a separate clade (21 isolates in that study) were likely to be other
*Capnocytophaga* species or subspecies of *C. canimorsus* [14]. After our phylogenetic analysis, we
obtained consistent results and considered that these strains were very likely to be other
*Capnocytophaga* species. In addition, in those strains were highly homogeneous with others but had
an only single deletion at the expected RE position, we speculated that there may be
[revised manuscript text omitted]

- 22. Gastra W, and Lipman LJ. *Capnocytophaga canimorsus*. *Vet Microbiol.* 2010;140:339-346.
- 23. Westwell AJ, Kerr K, Spencer MB, and Hutchinson DN. DF-2 infection. *BMJ.*
1989;298:116-117.
- 24. Renzi F, Manfredi P, Dol M, Fu J, Vincent S, and Cornelis GR. Glycan-foraging systems
reveal the adaptation of *Capnocytophaga canimorsus* to the dog mouth. *mBio.*
2015;6:e02507.
- 25. Peier AM, Moqrich A, Hergarden AC, Reeve AJ, Andersson DA, Story GM, et al. A TRP
channel that senses cold stimuli and menthol. *Cell.* 2002;108:705-715.
- 26. Nogueira BS, Martini AC, Menezes IG, Souza RL, Maruyama FH, Rodrigues JY, et al.
Detection of *Capnocytophaga canimorsus* and *Capnocytophaga cynodegmi* in dogs with
periodontal disease of Brazil. *Research, Society and Development.*
2021;10:e245101321146.
- 27. Tabatabaei M, and Rohani HR. Polymerase Chain Reaction Detection of *Capnocytophaga*
*canimorsus* and *Capnocytophaga cynodegmi* as the Emerging Zoonosis. *Avicenna J Clin*
*Microbiol Infect.* 2019;6:21-25.

**Figure Captions**

Figure 1. Phylogenetic tree and molecular typing results according to 16S rRNA gene sequences of
*Capnocytophaga* isolates and reference strains. A: Phylogenetic tree of all *Capnocytophaga* isolates
and reference strains. Reference strains are indicated by filled circles. Published sequences are
presented as strain No., GenBank accession No., and source. B: Molecular typing results of each
isolate are depicted in a 16S rRNA phylogenetic tree. Black squares and white squares correspond
to strains that were positive and negative, respectively, in the catalase test, oxidase test, or *C.*
*canimorsus*-specific (CaR) PCR test. Molecular typing according to the nucleotide polymorphism
of species-specific PCR reverse primer binding sites is presented using a color code, with the key
shown in the left upper corner of the diagram. T: type strain; H: human isolate; D: canine isolate; C:
feline isolate.

Figure 2. Nucleotide diversity of genus- or species-specific PCR primer recognition sites among 50
*Capnocytophaga* isolates. Five types were subdivided according to nucleotide polymorphisms from
positions 479 to 483. A: Sequences aligned with genus-specific PCR forward primer; B: Sequences
aligned with genus-specific PCR reverse primer; C: Sequences aligned with species-specific PCR
reverse primer. Gray boxes indicate nucleotides that differ between the two reverse primers of the
species-specific PCR.

Figure 3. PpuMI and KpnI restriction patterns for the digestion of amplified 16S rRNA gene
products. DNA fragments were analyzed by 1.5% agarose gel electrophoresis. M: 100-bp marker;
Ccy1: *C. cynodegmi* belongs to RFLP pattern type 2; Ccy2: *C. cynodegmi* belongs to RFLP pattern
type 3; Cca: *C. canimorsus*; *C. sp.*: strain 62067N and 91462, suspected new *Capnocytophaga*
species belong to RFLP pattern type 4; P: undigested PCR-amplified 16S rRNA product.

Figure 4. Phylogenetic trees were constructed based on approximately 440 bp of the 16S rRNA
sequence of published *C. canimorsus* and *C. cynodegmi* strains. A: Published *C. canimorsus* strains.
Strains without the expected RE sequences clustered in a separate clade, except for two strains
(indicated by filled triangles). B: Published *C. cynodegmi* strains. Strains without the expected RE
sequences were clustered in a clade with close proximity to other clades. Filled dots indicate strains
without the expected RE sequence.

Supplementary Figure 1. Predicted RE PpuMI restriction sites in all published *C. canimorsus* strains.
Nucleotide polymorphisms occurred predominantly in the expected RE sequence at positions 279
and 300. Light grey boxes indicate the restriction sites. Dark grey boxes indicate diverse
nucleotides.

Supplementary Figure 2. Predicted RE KpnI restriction sites in all published *C. cynodegmi* strains.
The nucleotide polymorphism that occurred only in the expected RE sequence was at position 475
as a deletion. The light grey box indicates the RE restriction site. The dark grey box indicates
diverse nucleotides.

Table 1. Newly designed 16S rRNA PCR-RFLP for identifying the *Capnocytophaga* species and isolates obtained in this study

Pattern type	Restriction enzyme				Fragment sizes (bp)	Represented species	Number of isolates (species)
	cutting site*						
	PpuMI		KpnI				
286	300	478	1414				
1	√	√	-	√	14, 51, 286, 1114	C. canimorsus	1 (C. canimorsus)
2	-	-	√	-	478, 987	C. cynodegmi	6 (C. cynodegmi)
3	-	-	√	√	51, 478, 936	C. cynodegmi , C. stomatis	41 (C. cynodegmi)
4	-	-	-	√	51, 1414	C. felis , C. canis	2 suspected new species

*Position number according to L14637

B Tree scale: 0.01

Reverse primer

- Type A
- Type B
- Type C
- Type D
- Type E

RFLP type

- Type 1
- Type 2
- Type 3
- Type 4

A

Position number according to L14637

| Strain | 175 | 176 | 177 | 178 | 179 | 180 | 181 | 182 | 183 | 184 | 185 | 186 | 187 | 188 | 189 | 190 | 191 | 192 | 193 | 194 | 195 | 196 |
|---|---|---|---|---|---|---|---|---|---|---|---|---|---|---|---|---|---|---|---|---|---|---|
| Primer AS1 | T | A | G | T | A | T | T | G | T | T | T | G | G | T | G | G | C | A | T | C | A | C |
| Cca ATCC 35979 | . | . | . | . | . | . | . | . | . | . | . | . | . | . | . | . | . | . | . | . | . | . |
| Ccy ATCC 49044 | . | . | . | . | . | . | . | . | . | . | . | . | . | . | . | . | . | . | . | . | . | . |
| 90470 | . | . | . | . | . | . | . | . | . | . | . | . | . | . | . | . | . | . | . | . | . | . |
| 70091 | . | . | . | . | . | . | . | A | . | . | . | A | . | . | . | . | . | . | . | . | . | T |
| 20093 | . | . | . | . | . | . | . | A | . | . | . | . | . | . | . | . | . | . | . | . | . | T |
| 91462 | . | . | . | . | . | . | . | A | . | . | . | T | A | . | . | . | . | . | . | . | . | T |
| 62067N | . | . | . | . | . | . | . | A | . | . | . | T | . | . | . | . | . | . | . | . | . | T |

B

Position number according to L14637

| Strain | 71 | 72 | 73 | | | 74 | 75 | 76 | 77 | 78 | 79 | 80 | 81 | 82 | 83 | 84 | 85 | 86 | 87 | 88 | 89 | 90 |
|---|---|---|---|---|---|---|---|---|---|---|---|---|---|---|---|---|---|---|---|---|---|---|
| Primer CaL2 | G | T | A | - | - | G | A | G | T | G | C | T | T | C | G | G | C | A | C | T | T | G |
| Cca ATCC 35979 | . | . | . | . | . | . | . | . | . | . | . | . | . | . | . | . | . | . | . | . | . | . |
| Ccy ATCC 49044 | . | . | . | . | . | . | . | . | . | . | . | . | . | . | . | . | . | . | . | . | . | . |
| 70091 | . | . | . | . | . | . | . | . | . | . | . | . | . | . | A | . | . | . | . | . | . | . |
| 98043 | . | . | . | . | . | . | . | . | . | . | . | . | . | . | . | . | . | G | . | . | . | . |
| WSCS300 | . | . | . | . | . | . | . | . | . | . | T | . | . | . | . | . | . | . | . | . | . | . |
| 60701 | . | . | . | . | . | . | G | . | . | . | . | . | . | . | . | . | . | . | G | . | . | . |
| 20093 | . | . | . | . | . | . | G | . | . | . | . | . | . | . | . | . | . | . | . | . | . | . |
| 91462 | . | . | . | A | G | . | . | . | A | . | . | . | . | G | C | T | . | T | . | C | C | T |
| 62067N | . | . | . | . | . | A | G | C | . | A | . | . | . | . | . | . | T | . | G | C | C | T |

C

Position number according to L14637

| Group | 479 | 480 | 481 | 482 | 483 | 484 | 485 | 486 | 487 | 488 | 489 | 490 | 491 | 492 | 493 | 494 | 495 | 496 | 497 |
|---|---|---|---|---|---|---|---|---|---|---|---|---|---|---|---|---|---|---|---|
| Primer CyR | C | A | T | A | C | G | A | A | T | A | A | G | C | A | T | C | G | G | C |
| Primer CaR | T | G | . | . | T | . | . | . | . | . | . | . | . | . | . | . | . | . | . |
| Ccy ATCC 49044 | . | . | . | . | . | . | . | . | . | . | . | . | . | . | . | . | . | . | . |
| Cca ATCC 35979 | T | G | . | . | T | . | . | . | . | . | . | . | . | . | . | . | . | . | . |
| Type | | | | | | | | | | | | | | | | | | | |
| Type A | . | . | . | . | . | . | . | . | . | . | . | . | . | . | . | . | . | . | . |
| Type B | . | G | . | . | . | . | . | . | . | . | . | . | . | . | . | . | . | . | . |
| Type C | . | . | . | G | . | . | . | . | . | . | . | . | . | . | . | . | . | . | . |
| Type D | T | G | . | . | T | . | . | . | . | . | . | . | . | . | . | . | . | . | . |
| Type E | T | C | . | G | . | . | . | . | . | . | . | . | . | . | . | . | . | . | . |

M

Ccy1

Ccy2

Cca

C. sp.

P

1500 bp

1000 bp

500 bp

300 bp

0.010

0.020

Spectrum02916-22

The manuscript title “A Novel 16S rRNA PCR-Restriction Fragment Length Polymorphism Assay to Accurately Distinguish Zoonotic *Capnocytophaga canimorsus* and *C. cynodegmi*” is about the new molecular tool for the epidemiology study of *Capnocytophaga* in small animals and for the rapid diagnosis of human *C. canimorsus* infections. The manuscript is good written but needs improve in the experimental studies and lack of innovative points. I have some comments to take in consideration to the authors:

1. The main problem of this work is the lack of the innovative technique.
2. If the study is about detection of the *Capnocytophaga canimorsus* and *C. cynodegmi*. Why the authors analyzed the Virulence factor analysis?
3. There are more studies published about molecular techniques about the detection of these pathogens? Include table about it.
4. 4. Add advantages and disadvantages of using of this technique.

Reviewer comments response

Comments from Reviewer 2:

- Comment 1: *The conventional isolation and identification methods could be explained more fluent.*

Response: Thank you for your comment. We have revised the isolation and identification procedures in lines 102-103 and 107-109 for clarity.

- Comment 2: *There are some deficiencies in terms of microbiological terminology.*

Response: Thank you for your comment. The deficiencies in microbiological terminology were revised in lines 48, 50, 68, and 98.

- Comment 3: *The authors can find the revision recommendations defined on the manuscript uploaded to the manuscript central.*

Response: Thank you for your revision recommendations. We did perform biochemical tests, but did not initially include them in our paper due to concerns about lengths. The manipulation, result, and discussion of the biochemical tests have now been added in lines 110-118, 173, 197-203, and 350-364, and Figure 1B. The minor residual revisions have been added to lines 93 and 121.

Comments from Reviewer 4:

- Comment 1: *The main problem of this work is the lack of the innovative technique.*

Response: Thank you for your comment. This study aimed to improve the accuracy of differentiating *C. canimorsus* from *C. cynodegmi* using a convenient method with low equipment requirements, that can hopefully be applied in laboratories when surveying *Capnocytophaga*. The discovery of nucleotide polymorphisms affects the species-specific PCR accuracy, and identifying an appropriate restriction enzyme that efficiently differentiates between these two *Capnocytophaga* species is critical for epidemiological research. We hope that this response clarifies the innovation component behind our study.

- Comment 2: *If the study is about detection of the Capnocytophaga canimorsus and C. cynodegmi. Why the authors analyzed the Virulence factor analysis?*

Response: Thank you for your question. Oxidase and catalase tests were part of the biochemical tests. Suzuki performed MLST analysis and found that the virulence of *C. canis* strains in humans is associated with oxidase activity (Suzuki et al, 2018). However, this association remains to be validated in *C. cynodegmi*. We have changed the “virulence factor analysis” section title to “biochemical tests”, and added some important biochemical results in lines 110-118, 173, and 197-203, and Figure 1B. The results also highlight that *C.*

cynodegmi strains had variable results from oxidase testing, and were not always positive.

- Comment 3: *There are more studies published about molecular techniques about the detection of these pathogens? Include table about it.*

Response: Thank you for your advice. Table 2 was constructed to demonstrate the techniques used to detect *C. canimorsus*- and *C. cynodegmi*-related species. Lines 277-278 have been added to the discussion section as well.

- Comment 4: *Add advantages and disadvantages of using of this technique.*

Response: Thank you for your comment. The advantages and disadvantages of this technique were added in lines 303-307.

Comments from Reviewer 5:

- Comment 1: *Figure 1A is a phylogenetic tree based on 16S rRNA gene sequences. Figure 1B uses the same phylogenetic tree, based on 16S rRNA gene sequences, and adds the molecular and phenotypic testing results in color-coded boxes. How should these two trees be interpreted differently?*

Response: Thank you for your question. Figure 1A emphasized the phylogenetic distance and clade variation between *Capnocytophaga* species. According to the reviewer's comments, we have made revisions in Figure 1B. Figure 1B now highlights the different phenotype characteristics between *C. canimorsus* and *C. cynodegmi*, and the impacts of reverse primer sequence type on *C. canimorsus*-specific PCR and RFLP typing among our isolates.

- Comment 2: *How do A-E type strains relate to 1-4 type strains?*

Response: Thank you for the important question. There was a color annotation error in the legend of Fig. 1B regarding reverse primer types C and E. We have corrected this error and also added carbohydrate utilization results according to the reviewer's comments. Based on the KpnI cleavage site, sequences with a "C" nucleotide at position 479 can be cleaved by KpnI at position 480 (Fig. 2C). Therefore, reverse primer types A, B, and C correspond to RFLP type 2 or 3, while types D and E correspond to RFLP type 1 or 4 (Fig. 1B). This has been added to lines 242-245.

- Minor comments:

1. *Lines 54-55: C. canimorsus usually causes systemic infections in immunocompromised individuals or compared to other Capnocytophaga species?*

Response: Thank you for your revision. The sentence was corrected to "Compared to other species, *C. canimorsus* usually causes systemic infections, including cellulitis, meningitis, and sepsis." in lines 54-55.

2. *Line 59: What is meant by "potential hazards"? Is this referring to virulence factors of zoonotic Capnocytophaga in the canine oral cavity?*

Response: Thank you for your question. We were trying to convey that zoonotic *Capnocytophaga* may pose a potential hazard to humans. *Capnocytophaga* bacteria are virtually non-pathogenic to small animals. This sentence has been revised (lines 60-61).

3. *Lines 137-138: This is not a full sentence.*

Response: Thank you for pointing out this error. We have revised the sentence as “The results from the aforementioned 16S rRNA gene sequence alignment of the isolates and reference strains indicated differences between *C. canimorsus* and *C. cynodegmi*” in lines 145-146.

4. *Line 171: The presence of Capnocytophaga may represent infection or colonization as the organism is part of the normal oral microbiota.*

Response: Thank you for your revision. The sentence was revised as “The prevalence of *Capnocytophaga* sp. in the canine oral cavity was 51% (49/98).” in lines 177-178.

5. *Lines 190-193: From a clinical microbiology perspective, most strains of C. canimorsus and cynodegmi are predicted to be oxidase and catalase positive in phenotypic identification testing. The significance of these virulence factors, or the lack of virulence factors for strain 62067N, is not discussed any further.*

Response: Thank you for your comment. Indeed, oxidase and catalase tests were mostly used for species differentiation in previous *Capnocytophaga* studies. However, Suzuki used MSLT analysis and found that the virulence of *C. canis* strains in humans is associated with oxidase activity (Suzuki et al, 2018). Whether this association also applies *C. cynodegmi* requires further validation. Thus, we revised “virulence factors” to “biochemical tests” in lines 110-118. The discussion of the possible virulence of oxidase in *C. cynodegmi* was added to lines 359-364.

6. *Line 302: Differentiating, not differencing.*

Response: Thank you for your revision. The word was revised as recommended in line 324.

7. *Line 465: Please amend figure legend to reflect that Cca should demonstrate the type 1 pattern (14, 51, 286, and 1114 bp) and that only the 286 and 1114 bp fragments are shown.*

Response: Thank you for your advice. The figure legend was revised as recommended in lines 502-505.

February 9, 2023

Dr. Wei-Yau SHIA
National Chung Hsing University
Veterinary Medicine
145, XingDa Rd.
Taichung 402
Taiwan

Re: Spectrum02916-22R1 (A Novel 16S rRNA PCR-Restriction Fragment Length Polymorphism Assay to Accurately Distinguish Zoonotic *Capnocytophaga canimorsus* and *Capnocytophaga cynodegmi*)

Dear Dr. Wei-Yau SHIA:

Please revise the manuscript according to the concerns of reviewer #3 and return it as soon as possible. Failure to address reviewer #3's concerns will lead to rejection.

Link Not Available

Sincerely,

John Osei Sekyere

Journals Department
Reviewer comments:

Reviewer #1 (Comments for the Author):

NA

Reviewer #3 (Comments for the Author):

Thank you for addressing all my comments and suggestions.

Reviewer #5 (Comments for the Author):

Comments for the Author:

The manuscript by Lai, et al. has been improved by the revisions and clarifications however, there are some remaining questions and concerns. Please see my major and minor comments below.

Major comments:

In my opinion, there is missing rationale for this work that makes this study a little challenging to follow. In the methods section, 16S rRNA sequencing was performed to validate isolates not identified by genus- and species-specific PCR. However, in the results section, the authors first describe 16S rRNA sequencing, BLAST and phylogenetic analyses for identification of Capnocytophaga. The authors then compared the sequencing identification (gold standard) to biochemical identification, their novel RFLP analysis, and the more traditional genus- and species-specific PCR to determine if the RFLP could more accurately identify species of Capnocytophaga. The failure of genus- and species-specific PCR to identify and differentiate Capnocytophaga species (with the final identification coming from the 16S rRNA sequencing result) is what prompted the need to develop PCR-RFLP, an approach that the authors argue is an accurate, cost effective and less labor-intensive approach. Based on the data presented here with sequencing results presented first as a standalone result, it is difficult to understand the aim and innovation of the work.

Minor comments:

Line 52: "...common transmission pathway to humans is via..."

Line 106: "Amplified for" not "amplified from".

Line 108: Please clarify 16S rRNA confirmation if different from what is described in section 2.4 below. Is 16S rRNA confirmation sequencing of the entire 16S rRNA gene or as described in sections 2.4 and 2.5? The authors could add a reference to section 2.4 if it is the same.

Line 199: "...16S rRNA result."

Lines 199-201: This sentence needs to be clarified. Do the authors mean "18% of *C. cynodegmi* strains were misidentified by biochemical methods as *C. canimorsus* or other non-Capnocytophaga species"?

Figure 1: Recommend changing the figure legend colors to colors that are high contrast and accessible for color-blind audiences (e.g., red-blue, blue-green, red-purple).

Line 305: "Possibly" not "possible".

Staff Comments:

Preparing Revision Guidelines

Please return the manuscript within 60 days; if you cannot complete the modification within this time period, please contact me. If you do not wish to modify the manuscript and prefer to submit it to another journal, please notify me of your decision immediately so that the manuscript may be formally withdrawn from consideration by Microbiology Spectrum.

If your manuscript is accepted for publication, you will be contacted separately about payment when the proofs are issued; please follow the instructions in that e-mail. Arrangements for payment must be made before your article is published. For a

complete list of **Publication Fees**, including supplemental material costs, please visit our website.

Reviewer comment responses

Comments from Reviewer 5:

- Comment 1: In my opinion, there is missing rationale for this work that makes this study a little challenging to follow. In the methods section, 16S rRNA sequencing was performed to validate isolates not identified by genus- and species-specific PCR. However, in the results section, the authors first describe 16S rRNA sequencing, BLAST and phylogenetic analyses for identification of Capnocytophaga. The authors then compared the sequencing identification (gold standard) to biochemical identification, their novel RFLP analysis, and the more traditional genus- and species-specific PCR to determine if the RFLP could more accurately identify species of Capnocytophaga. The failure of genus- and species-specific PCR to identify and differentiate Capnocytophaga species (with the final identification coming from the 16S rRNA sequencing result) is what prompted the need to develop PCR-RFLP, an approach that the authors argue is an accurate, cost effective and less labor-intensive approach. Based on the data presented here with sequencing results presented first as a standalone result, it is difficult to understand the aim and innovation of the work.

1.1. Response: Thank you for this comment. To highlight the importance of our designed method on epidemiology prevalence, we have made some changes to the Materials and Methods and Results. First, we merged the identification and phylogenetic analysis into one section (Section 2.2) in the Materials and Methods (lines 101–134). We now start the Results with a section called “*Prevalence investigated using conventional methods—species-specific PCR and 16S rRNA sequences* (Section 3.1)”, followed by “*Effect of site-specific sequence polymorphism on genus- and species-specific PCR and the phylogenetic tree* (Section 3.2)”, and have a separate “Biochemical identification” section (Section 3.3) (lines 180–237). Finally, we present our novel PCR-RFLP method (Section 3.4) and its verification (Section 3.5). Our intention is to first present the results of previously designed species-specific PCR to illustrate the problems associated with these methods in *Capnocytophaga* epidemiological surveys. Next, we investigate the reasons for the problem, and finally, we solve the problem with the newly designed method. We hope that the new order of the results makes them easier for the reader to follow.

- Minor comments:

1. Line 52: "...common transmission pathway to humans is via..."

Response: Thank you for this revision. We have revised the sentence as suggested in lines 50–51.

2. Line 106: "Amplified for" not "amplified from".
Response: Thank you for this revision. The sentence has been revised in lines 109–110.
3. Line 108: Please clarify 16S rRNA confirmation if different from what is described in section 2.4 below. Is 16S rRNA confirmation sequencing of the entire 16S rRNA gene or as described in sections 2.4 and 2.5? The authors could add a reference to section 2.4 if it is the same. The 16S rRNA confirmation sequencing was the same as described in section 2.4.
Response: Thank you for your question. The 16S rRNA confirmation is the same as that described in Section 2.4. We have merged these sections for better understanding (lines 110–116).
4. Line 199: "...16S rRNA result."
Response: Thank you for this comment. The sentence has been revised as "...16S rRNA sequencing results" in lines 229–230.
5. Lines 199-201: This sentence needs to be clarified. Do the authors mean "18% of *C. cynodegmi* strains were misidentified by biochemical methods as *C. canimorsus* or other non-*Capnocytophaga* species"?
Response: Thank you for this question. The sentence has been revised as “In contrast, 18% (8/44) of the *C. cynodegmi* strains were misidentified as *C. canimorsus* (five strains) or other non-*Capnocytophaga* species (three strains)” in lines 230–231.
6. Figure 1: Recommend changing the figure legend colors to colors that are high contrast and accessible for color-blind audiences (e.g., red-blue, blue-green, red-purple).
Response: Thank you for this comment. The figure legend colors have been revised (Figure 1B).
7. Line 305: "Possibly" not "possible".
Response: Thank you for this comment. The word has been revised in line 321.

May 4, 2023

Dr. Wei-Yau SHIA
National Chung Hsing University
Veterinary Medicine
145, XingDa Rd.
Taichung 402
Taiwan

Re: Spectrum02916-22R2 (A Novel 16S rRNA PCR-Restriction Fragment Length Polymorphism Assay to Accurately Distinguish Zoonotic *Capnocytophaga canimorsus* and *Capnocytophaga cynodegmi*)

Dear Dr. Wei-Yau SHIA:

Your manuscript has been accepted, and I am forwarding it to the ASM Journals Department for publication. You will be notified when your proofs are ready to be viewed.

Sincerely,

John Osei Sekyere
Editor, Microbiology Spectrum